# Dynamic Sparse Network for Time Series Classification: Learning What to "See"

**Qiao Xiao**[1,*]**, Boqian Wu**[2,*]**, Yu Zhang** [3,4]**, Shiwei Liu**[5,1]**, Mykola Pechenizkiy**[1]**,**
**Elena Mocanu**[2]**, Decebal Constantin Mocanu** [1,2]
[1] Eindhoven University of Technology, [2] University of Twente
[3] Southern University of Science and Technology
[4] Peng Cheng Laboratory, [5] University of Texas at Austin
`{q.xiao,m.pechenizkiy}@tue.nl`
`{b.wu,e.mocanu,d.c.mocanu}@utwente.nl`
`yu.zhang.ust@gmail.com`
`shiwei.liu@austin.utexas.edu`

## Abstract

The receptive field (RF), which determines the region of time series to be "seen" and used, is critical to improve the performance for time series classification (TSC). However, the variation of signal scales across and within time series data, makes it challenging to decide on proper RF sizes for TSC. In this paper, we propose a dynamic sparse network (DSN) with sparse connections for TSC, which can learn to cover various RF without cumbersome hyper-parameters tuning. The kernels in each sparse layer are sparse and can be explored under the constraint regions by dynamic sparse training, which makes it possible to reduce the resource cost. The experimental results show that the proposed DSN model can achieve state-of-art performance on both univariate and multivariate TSC datasets with less than 50% computational cost compared with recent baseline methods, opening the path towards more accurate resource-aware methods for time series analyses. Our code is publicly available at: `https://github.com/QiaoXiao7282/DSN`.

## 1 Introduction

Time series classification (TSC) as an important research topic in the data mining communities [14], has a wide range of applications from health monitoring [24, 16], and public security [58], to grid energy [41], and remote sensing [43]. In the last decade, neural networks, especially deep neural networks, have achieved competitive or even better performance than traditional TSC approaches (e.g. DTW [46], BOSS [47], COTE [1]) in many cases [56, 14].

However, how to discover and exploit the various scaled signals hidden in time series (TS) is still a significant challenge for TSC tasks [52, 36, 21]. One main reason lies in several variances, like sampling rate and record length, which are natural during time-series collection [47, 31, 5]. Furthermore, the amplitude offset, warping and occlusion of data points are unavoidable (see Figure 1). So determining the optimal scales for feature extraction is difficult but significant in the TSC task. One main solution is to cover as many receptive fields (RF) scales as possible, so it does not ignore any useful signals from the time series inputs [52, 7, 8, 25].

Grounding on the motivation that multiple kernels in combination can capture discriminative patterns despite complex warping, Fawaz et al. [15] tried to ensemble convolutional neural networks with a large number and variety of kernels. Tang et al. [53] have stacked several OS-CNN blocks, each of

---

*[*]Equal contribution.*

which consisted of a list of kernel sizes to cover all scales of RF and achieve better performance on several TSC tasks. In Multi-scale Convolutional Neural Network (MCNN) [4], a grid search was applied to find suitable kernel sizes. Without elaborate setting and fine-grid search, Dempster et al. [7] transformed time series by using random convolutional kernels and then trained a time series classifier. Even though the use of kernels with various sizes can help to extract latent hierarchical features of multiple resolutions, it tends to cause a huge computational problem and overfits the datasets when a limited amount of training data is available [25]. To address this problem, dilated convolution [59] has been used to reduce the computational cost while keeping the receptive field size. Franceschi et al. [17] presented an unsupervised learning model with convolutional kernels for time series feature transformation, in which the dilation factors of kernels increased exponentially layer by layer. To date, there is still an unresolved challenge with hyperparameter selections (e.g. kernel sizes and dilated factors) to capture patterns at different scales. This raises a foundational question: ***Can we design a scalable method for TSC that achieves a trade-off between computation and performance without cumbersome hyperparameter tuning?***

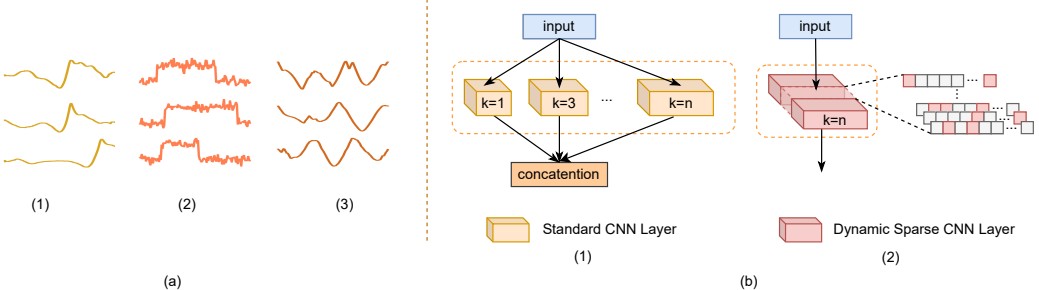

Figure 1: (a) Toy example illustrates the various time scales characteristic of TS data. Data in (1), (2), and (3) are sampled from three different TS datasets. The sequences with the same color represent the same class. We can find that several sources of variances like phase, warping and offset are common in TS data. (b) The comparison between other methods [53, 25] (shown in b1) and our proposed DSN (shown in b2) for TSC. To cover various RF, DSN does not need to be configured with a series of kernel sizes $k$.

Inspired by sparse neural network models which can match the performance of dense counterparts with fewer connections [40, 35, 33, 60], we propose a dynamic sparse network (DSN) that is composed of CNN layers with large but dynamic sparse kernel for TSC tasks, which can automatically learn sparse connections in terms of covering various RF. The proposed DSN is trained by a dynamic sparse training (DST) strategy, which makes it possible to reduce the computational cost (e.g. floating-point operations (FLOPs)) for training and inference as well. Different from the traditional dynamic sparse training methods, which explore the connections in a layer-wise manner [40, 34, 13] and can hardly cover smaller RF, we propose a more fine-grained sparse training strategy for TSC. Specifically, the CNN kernels in each layer are split into several groups, each of which is sparse and can be explored under a constraint region during sparse training. In this way, each layer can extract features in a more scalable and diverse way, and then by cascading, the model can cover more various scales of RF. To sum up, our main contributions are:

- We propose a novel dynamic sparse CNN layer with various effective neighbour RF, which can learn to "see" selectively and adaptively for more accurate TSC.

- By stacking the above dynamic sparse CNN layers, we introduce a novel DSN model natively for TSC, which is trained by DST to reduce computational and memory cost.

- To bridge the gap between resource awareness and accuracy, we propose a more fine-grained training strategy for DSN, which can more easily cover diverse effective RF for TSC.

- We perform extensive empirical evaluation on both univariate and multivariate TSC tasks, showing that DSN achieves superior performance in terms of accuracy and resource cost.

## 2 Related Work

### 2.1 Time Series Classification

In the last decade, the success of deep learning encourages researchers to explore and extend its application on TSC tasks [28, 14, 56, 23, 29]. For univariate TSC, deep learning-based models attempt to directly transform raw time series data into low-dimensional feature representation via convolutional or recurrent networks [25, 45, 42]. For multivariate TSC, LSTM and attention layers are often stacked with CNN layers to extract complementary features from TS data [26, 62].

Recently, many works have attempted to extract features in a more scalable manner as TS data is unavoidably composed of signals on various scales [47, 5, 52]. One main solution is to configure suitable kernels of various sizes such that they increase the probability of capturing the proper scales [52, 53, 25]. Rocket-based methods [7, 8, 9] aim to use random kernels with several sizes and dilation factors to cover diverse RF for TSC. Different from these works, our proposed dynamic sparse CNN layer can adaptively learn the various RF with deformable dilation factors and trade-off both computation and performance without cumbersome hyperparameter tuning.

### 2.2 Sparse Training

Recently, the lottery ticket hypothesis has shown that it is possible to train a sparse sub-network to match the performance of its dense counterpart with less computational cost [18]. Rather than iteratively pruning from a dense network [19, 63, 18], recent works try to find an initial mask with one-shot pruning based on gradient information during training [30, 55]. After pruning, the topology of the neural network will be fixed during training. However, this kind of models can hardly match the accuracy achieved by their dense counterparts [38, 55, 51].

Introduced as a new training paradigm before the lottery ticket hypothesis, DST starts from a sparse network and allows the sparse connectivity to be evolved dynamically with a fixed number of parameters throughout training [40, 13, 35, 60, 10]. Nowadays, DST attracts increasing attentions from other research fields such as reinforcement learning [49] and continual learning [48], due to its potential of outperforming dense neural networks training [40, 60, 35]. Different from the traditional DST methods, our proposed DSN is trained with a fine-grained sparse training strategy rather than a traditional layer-wise manner to capture more diverse RF for TSC.

### 2.3 Adaptive Receptive Field

The RF, which can be adaptively changed during the training, has proven to be effective in many domains [54, 44, 57]. The adaptive RF can usually be captured by learning the optimal kernel sizes or the masks of kernels during the course of training [50, 20, 57]. However, neither the kernels nor the masks are sparse, which may cause the huge computational problem when larger RF is needed. Different from these methods, our dynamic sparse CNN layers in DSN can be trained with DST, which can learn to capture variable RF with deformable dilation factors. What's more, the kernels are always sparse during training and inference, which makes it possible to reduce the computational cost.

## 3 The Proposed Model

### 3.1 Problem Definition

**Definition 1.** *(Time Series Classification (TSC)). Given a TS instance* $\mathbf{X} = \{\mathbf{X}_1, \ldots, \mathbf{X}_n\} \in \mathbb{R}^{n \times m}$, *where* $m$ *denotes the number of variates and* $n$ *denotes the number of time steps, TSC aims to accurately predict the class label* $y \in \{1, \ldots, c\}$ *from* $c$ *classes. When* $m$ *equals 1, TSC is univariate and otherwise it is multivariate.*

**Definition 2.** *(Time Series (TS) Training Set). A training set* $\mathcal{D} = \{(\mathbf{X}^{(1)}, y^{(1)}), \ldots, (\mathbf{X}^{(N)}, y^{(N)})\}$ *consists of* $N$ *time series instances, where* $\mathbf{X}^{(i)} \in \mathbb{R}^{n \times m}$ *could either be a univariate or multivariate time series instance with its corresponding label* $y^{(i)} \in \{1, \ldots, c\}$.

Note that, in our case, all instances have the same number of time steps in a TS dataset. Without the loss of generality, given a training set, we aim to train a CNN classifier with adaptive and various RF with a low resource cost (e.g. memory and computation) for the TSC task.

## 3.2 Dynamic Sparse CNN Layer with Adaptive Receptive Field

A straightforward strategy to cover various RF is to apply multi-sized kernels in each CNN layer, but it has several limitations. Firstly, TS instances from different TS datasets do not have the same length and cycles [2] in most cases, making it difficult to set a fixed kernel configuration for all the datasets even with prior knowledge. Secondly, obtaining a large receptive field commonly requires a large kernel or stacking more layers, which introduces more parameters and thus increases storage and computational costs.

To tackle those challenges, the proposed dynamic sparse CNN layer possesses large but sparse kernels, which are learnable to capture adaptive RF. Specifically, given the input feature map $x^l \in \mathbb{R}^{c_{l-1} \times h \times w}$ in the $l$th layer ($h$ equals 1 for univariate TSC and $c_{l-1}$ is the number of input channels) , and kernel weights $\Theta^l \in \mathbb{R}^{c_{l-1} \times c_l \times 1 \times k}$ ($k$ is the kernel size and $c_l$ is the number of output channels), the convolution with stride 1 and padding in our proposed dynamic sparse CNN layer is formulated as

$$\mathbf{O}_j = \sum_{0 < i \leq c_{l-1}, i \in \mathbb{Z}} \left( \mathbf{I}^l(\Theta^l)_{i,j} \odot \Theta^l_{i,j} \right) \cdot x^l_i, \tag{1}$$

where $\mathbf{O}_j \in \mathbb{R}^{h \times w}$ denotes the output feature representation at the $j$-th output channel, $\mathbb{Z}$ denotes the set of integers, $\mathbf{I}^l(\cdot) : \mathbb{R}^{c_{l-1} \times c_l \times 1 \times k} \rightarrow \{0,1\}^{c_{l-1} \times c_l \times 1 \times k}$ is an indicator function, $\mathbf{I}^l(\Theta^l)_{i,j}$ indicates activated weights for $\Theta^l_{i,j} \in \mathbb{R}^{1 \times k}$ which is the kernel in the $(i,j)$-th channel, $\odot$ denotes the element-wise product, and $\cdot$ denotes the convolution operator. The indicator function $\mathbf{I}^l(\cdot)$, which is learned during the training of the proposed DSN, satisfies that $\|\mathbf{I}^l(\cdot)\|_0 \leq (1-S)N_l$ where $0 \leq S < 1$ is the sparsity ratio, $\|.\|_0$ denotes the $L_0$ norm and $N_l = c_{l-1} \times c_l \times 1 \times k$. When $S > 0$, the kernel is sparse, and we can use a large $k$ in the dynamic sparse CNN layer to obtain a large RF with reduced computational cost.

**Remark.** *(Effective Neighbour Receptive Field (eNRF) size in the dynamic sparse CNN layer). The receptive field is defined as the region in the input that the feature of a CNN model is looking at. We defined the Neighbour Receptive Field (NRF) as the region considered by each feature in the successive layer. Specifically, the NRF size is equivalent to the kernel size in the standard CNN layer (Considering the case that dilation equals to 1). Differently, the NRF size of the proposed dynamic sparse CNN layer is smaller than the kernel size, when the first or last weight in a kernel is not activated, e.g., $\exists i \in \{1, \ldots, c_{l-1}\}$ and $j \in \{1, \ldots, c_l\}$, such that $I^l(\Theta^l)_{i,j,1,1} = 0$ or $I^l(\Theta^l)_{i,j,1,k} = 0$. As illustrated in Figure 2, we defined the eNRF size $f^l_{i,j}$ of the kernel $\Theta^l_{i,j}$ as the distance between the first activated weight and last activated weight in the $l$-th CNN layer as*

$$f^l_{i,j} := \begin{cases} \max(Ind^l_{i,j}) - \min(Ind^l_{i,j}) + 1, & \text{if } Ind^l_{i,j} \neq \varnothing \\ 0, & \text{otherwise} \end{cases}, \tag{2}$$

*where $Ind^l_{i,j}$ denotes the set of indices corresponding to non-zero weights in the kernel $\Theta^l_{i,j}$. Obviously we have $0 \leq f^l_{i,j} \leq k$.*

The eNRF size set of $l$-th dynamic sparse CNN layer is denoted by $\mathbb{F}^{(l)}$, which satisfies that $0 \leq \min(\mathbb{F}^{(l)})$ and $\max(\mathbb{F}^{(l)}) \leq k$. Take the case in the Figure 2 as an simple example for $l$-th layer, then $\mathbb{F}^{(l)} = \{0, 1, 2, 3, 4, 5, 6\}$. Each dynamic sparse CNN layer is likely to cover various eNRF sizes ranging from 1 to $k$. When global information is expected, $\mathbf{I}^l(\cdot)$ could activate weights more dispersed to get a larger eNRF and exploit the input features selectively. To capture the local context, a smaller eNRF is expected, thus the activated weights tend to be

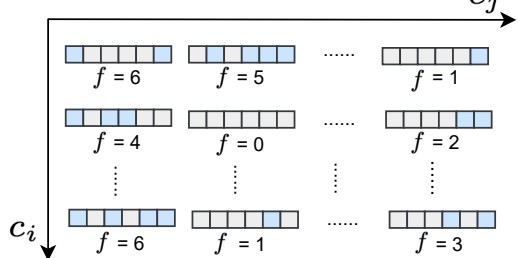

Figure 2: Illustrations for the eNRF size. The weights with blue color are activated.

---

[2] The term 'cycle' means that the data exhibit rising and falling that are not of a fixed frequency.

concentrated. By taking $k = 5$ as an example, $\mathbf{I}^l(\Theta^l)_{i,j}$ may be [1, 0, 1, 0, 1] for global context, while it may be [0, 0, 1, 1, 0] for local context. In this way, the eNRF can be adaptively adjusted.

## 3.3 Architecture in DSN

The proposed DSN model consists of three sparse CNN modules, each of which composes of a dynamic sparse CNN layer and a $1 \times 1$ CNN layer. Following the stacked sparse CNN modules, there is an additional dynamic sparse CNN layer, two adaptive average pooling layers, and one $1 \times 1$ convolution layer acting as the classifier in the DSN model. The overall architecture is shown in Figure 3.

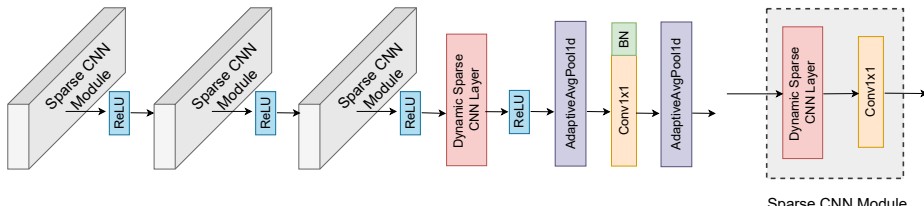

Figure 3: The proposed DSN model (left), which includes several sparse CNN modules (right) followed by a dynamic sparse CNN layer, two adaptive average pooling layers, and one $1 \times 1$ convolution layer.

The set of eNRF sizes $\mathbb{S}^{(l)}$ in the $l$th sparse CNN module is equal to that in the dynamic sparse CNN layer, since the eNRF size of $1 \times 1$ convolution is constantly equal to 1. $\mathbb{S}^{(l)}$ satisfies that $0 \leq \min(\mathbb{S}^{(l)})$ and $\max(\mathbb{S}^{(l)}) \leq k$. Then, the set of eNRF sizes for the three successively stacked sparse CNN modules can be described by $\mathbb{RF}$ as

$$\mathbb{RF} = \left\{ \max(0, s^{(1)} + s^{(2)} + s^{(3)} - 2) \mid s^{(i)} \in \mathbb{S}^{(i)}, i \in \{1, 2, 3\} \right\}. \tag{3}$$

According to Eq. (3), we can see that stacking multiple sparse CNN modules can increase the size of $\mathbb{RF}$ linearly, where $l$th dynamic sparse CNN layer increases by the size of $\mathbb{S}^{(l)}$. For simplicity, the kernel size in each dynamic sparse CNN layer is consistently set to $k$ in our study. Then, $\mathbb{RF}$ satisfies that $\max(\mathbb{RF}) \leq 3k - 2$ and $0 \leq \min(\mathbb{RF})$. Thus, a large $k$ in sparse CNN modules can increase the range of the eNRF sizes for stacked sparse CNN modules.

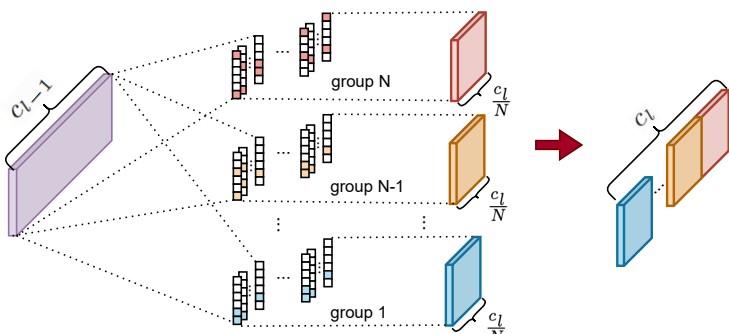

Figure 4: An illustration of the dynamic sparse CNN layer. In a dynamic sparse CNN layer, the kernels are split into $N$ groups, each of which is with sparse connections learned by dynamic sparse training across specific constraint regions. The connections with color indicate non-zero values.

## 3.4 Dynamic Sparse Training for DSN Model

In this section, we present the training strategy to discover the weights required to be activated in order to ensure well-performing RF. That is, we need to study how to update the indicator function $\mathbf{I}^l(\cdot)$ during the training of the proposed DSN.

We sparsely train the proposed DSN model from scratch to keep kernel sparse following the main idea of DST methods [40]. By design, during the training phase, the total number of activated weights must not exceed $N_l(1 - S)$. However, we observe that the tiny eNRF is hard to be captured by DST methods with a layer-wise exploration manner, namely activated weights are discovered layer by layer, especially when the sparsity ratio $S$ is undersized (more analysis in Section 4.5). Motivated by this observation, the kernels in each dynamic sparse CNN layer are divided into different groups, whose corresponding exploration regions are of different sizes as shown in Figure 4. Contrary to the DST methods, in DSN, the exploration of activated weights is separately performed in different kernel groups, which is a more fine-grained strategy. Specifically, the kernel weights $\Theta^l \in \mathbb{R}^{c_{l-1} \times c_l \times 1 \times k}$ in the $l$th layer are split into $N$ groups along the output chan-

**Algorithm 1** Dynamic Sparse Training for DSN Model

**Input:** Dataset $\mathcal{D}$, Network $f_\Theta$, Sparsity ratio $S$
    Exploration regions in layer $l$: $\{\mathcal{R}_1^l, ..., \mathcal{R}_N^l\}$
    Exploration schedule: $T, \Delta T, \alpha, f_{\text{decay}}$
1:  $\theta_i^l \leftarrow$ initialize activated weights in $\Theta_i^l$ using $S$ and $\mathcal{R}_i^l$
2:  **for** $t = 1$ **to** $T$ **do**
3:     Sample a batch $B_t \sim \mathcal{D}$
4:     $L_t = \sum_{i \sim B_t} L\left((f_\theta(x_i), y_i\right)$
5:     **if** $(t \mod \Delta T) == 0$ **then**
6:       **for** each layer $l = 1$ **to** $L$ **do**
7:         **for** each group $i = 1$ **to** $N$ **do**
8:           $u = n(\mathcal{R}_i^l) f_{\text{decay}}(t; \alpha, T)(1 - S)$
9:           $\mathbb{I}_{\text{prune}} = \text{ArgTopK}\left(-\left|\theta_i^l\right|, u\right)$
10:          $\mathbb{I}_{\text{grow}} = \text{RandomK}\left(\mathcal{R}_i^l \backslash \theta_i^l, u\right)$
11:          $\mathbf{I}^l(.) \leftarrow$ Update $\mathbf{I}^l(.)$ using $\mathbb{I}_{\text{prune}}$ and $\mathbb{I}_{\text{grow}}$
12:          $\theta_i^l \leftarrow$ Update activated weights $\mathbf{I}^l(\Theta_i^l) \odot \Theta_i^l$
13:         **end for**
14:       **end for**
15:     **else**
16:       $\theta = \theta - \alpha \nabla_\theta L_t$
17:     **end if**
18: **end for**

nel, that is, $\Theta_1^l, \ldots, \Theta_N^l \in \mathbb{R}^{c_{l-1} \times \frac{c_l}{N} \times 1 \times k}$ with corresponding exploration regions $\mathcal{R}_1^l, \ldots, \mathcal{R}_N^l$. The exploration region $\mathcal{R}_i^l$ in the $i$th group of the $l$th layer is defined as the first $\frac{i \times k}{N}$ positions of each kernel in this group. By taking $k = 6$ and $N = 3$ for example, activated weights in the first group are only within the first two positions of each kernel in $\Theta_1^l$, while they are within the whole kernel for the last group (as shown in Figure 4). In this way, the various eNRF can be covered, and exploration space can be reduced (more details can be found in Appendix A.1) to improve exploration efficiency.

Given the weight exploration regions $\mathcal{R}_1^l, \ldots, \mathcal{R}_N^l$ and the sparsity ratio $S$, we train the proposed DSN model as shown in Algorithm 1. The activated weights are explored within the exploration regions and updated every $\Delta_t$ iterations. The fraction of updated weight decays over time according the function $f_{\text{decay}}(t; \alpha, T)$, which follows the cosine annealing [11] as

$$f_{decay}(t; \alpha, T) = \frac{\alpha}{2}\left(1 + \cos\left(\frac{t\pi}{T}\right)\right), \tag{4}$$

where $\alpha$ is the initial fraction of activated weights updated, $t$ is the current training iteration, and $T$ is the number of training iterations. Thus, during the $t$th iteration, the number of updated activated weights in the $i$th group of the $l$th layer is $n(\mathcal{R}_i^l) f_{\text{decay}}(t; \alpha, T)(1 - S)$, where $n(\mathcal{R}_i^l)$ is the number of weights can be explored in region $R_i^l$. During the update of activated weights, we firstly prune the activated weights determined by $\text{ArgTopK}\left(-\left|\theta_i^l\right|, u\right)$ and then randomly grow new activated weights by $\text{RandomK}\left(\mathcal{R}_i^l \backslash \theta_i^l, u\right)$, where $\text{ArgTopK}(\mathbf{v}, u)$ gives the indices of top $u$ elements with larger values in a vector $\mathbf{v}$, $\text{RandomK}(\mathbf{v}, u)$ outputs the indices of random $u$ elements in a vector $\mathbf{v}$, and $\mathcal{R}_i^l \backslash \theta_i^l$ denotes the weights within $\mathcal{R}_i^l$ but except $\theta_i^l$.

The exploration of activated weights is straightforward. Firstly, pruning weights with small magnitudes is intuitive, because the contribution of weights with smaller magnitudes is insignificant or even negligible. Considering the pruning recoverability, we randomly regrow new weights with the same number as the pruned weights to achieve better activated weight exploration. In this way, the exploration of weights is dynamic and of plasticity, compared with the methods of pruning weights before and after training (more explanations about pruning and dynamic sparse training can be found in [22, 39, 32]).

## 4 Experiments

In this section, we evaluate the resource cost and accuracy of the proposed DSN model with recent baseline methods on both univariate and multivariate TSC data.

### 4.1 Datasets and Baselines

The details of each kind of dataset are as follows.

- **Univariate TS dataset from UCR 85 archive [3]**: This archive consists of 85 univariate TS datasets, which are collected from various domains (e.g. health monitoring and remote sensing), have distinguishable characteristics, and various levels of complexity. The number of instances in the training set varies from 16 to 8926, while their time step resolutions range from 24 to 2709.

- **Three multivariate TS datasets from UCI [12]**: The EEG2 dataset contains 1200 instances of 2 categories and 64 variates. The Human Activity Recognition (HAR) dataset consists of 10,299 instances with 6 categories, and the number of variates is 9. The Daily Sport dataset contains 9,120 instances of 19 categories, and its number of variates is 45. The processing of these datasets is the same as in [26].

**Univariate TSC baselines:** The performance of the following three baseline methods are reported, OS-CNN [52], InceptionTime [25], and ResNet[56], as they are widely used for univariate TSC. **Multivariate TSC baselines:** Similar with [62], the following three multivariate baselines are selected: OS-CNN, TapNet [62], MLSTM-FCN [26]. To avoid an unfair comparison, we let outside the ensemble methods. However, based on our best knowledge, our performance analysis accounts for most of state-of-the-art methods.

### 4.2 Experimental Settings and Implementation Details

For optimization, we use the Adam optimizer with an initial learning rate of $3 \times 10^{-4}$, cosine decayed to $10^{-4}$. Our model is trained for 1,000 epochs with a mini-batch size of 16. As in [25, 26], the best model, which corresponds to the minimum training loss, is used to evaluate performance over the testing sets. Inspired by [25], the kernel sizes $k$ in each dynamic sparse CNN layer are set to 39. The number of kernel groups $N$ in Algorithm 1 is set to 3, which helps to cover the small, medium, and large eNRF. Each setting is repeated five times, and the average results are reported.

### 4.3 Results and Analysis

We show the performance on both univariate and multivariate TSC benchmarks in Table 1-2. For univariate TSC, we can see that our method outperforms the baseline methods in most cases with a smaller number of parameters (e.g. Params [3]) and less computation cost (e.g. FLOPs) on UCR 85 archive datasets. For multivariate TSC, our proposed DSN method achieves better performance on EEG2 and HAR datasets, while for Daily Sport, MLSTM-FCN was with 0.46% more accurate. It is worth highlighting that the resource cost (i.e. Params and FLOPs) of our proposed DSN is much smaller than that of baseline methods. Due to the page limitation, the detailed results, including test accuracy and resource cost on each dataset of UCR 85 archive, can be found in Appendix C. More results on UCR 128 [6] and UEA 30 archive [2] can be found in Appendix B.9.

Table 1: Pairwise comparison of test accuracy (%) and mean resource cost (i.e. Params ($K$)↓ and FLOPs ($M$)↓) of our method and other univariate TS baseline methods on UCR 85 archive datasets. Note that all the accuracies in comparison are rounded to two decimal places.

| Archive | Methods | Baseline wins | **DSN** wins | Tie | Params | FLOPs |
|---|---|---|---|---|---|---|
| | ResNet | 29 | **53** | 3 | 479.20 | 803.38 |
| | InceptionTime | **40** | **40** | 5 | 389.34 | 325.49 |
| UCR 85 archive | OS-CNN | 33 | **49** | 3 | 262.14 | 200.12 |
| | **DSN (ours)** | - | - | - | **126.22** | **104.88** |

### 4.4 Sensitivity Analysis

**Effect of Sparsity Ratio:** From Table 3, we can find out how the sparsity ratio of dynamic sparse CNN layer affects the final test accuracy and the resource cost in multivariate TS datasets. Here,

---

[3]The amount of non-zero parameters in the model as in [49, 27, 30].

Table 2: Test accuracy (ACC(%)) and resource cost (i.e. Params $(K)\downarrow$ and FLOPs $(M)\downarrow$) of different models on three multivariate TS datasets from UCI.

| Methods | Daily Sport | | | EEG2 | | | HAR | | |
|---|---|---|---|---|---|---|---|---|---|
| | ACC | Params | FLOPs | ACC | Params | FLOPs | ACC | Params | FLOPs |
| TapNet | 99.56 | 1885.58 | 346.97 | 85.30 | 1369.21 | 780.19 | 93.33 | 1275.78 | 310.61 |
| MLSTM-FCN | **99.65** | 324.77 | 70.59 | 91.00 | 342.51 | 161.53 | 96.71 | 284.97 | 63.38 |
| OS-CNN | 99.30 | 317.05 | 78.01 | 93.03 | 320.34 | 162.04 | 96.37 | 280.39 | 70.98 |
| **DSN (ours)** | 99.19 | **163.68** | **40.48** | **99.10** | **162.14** | **82.31** | **96.82** | **160.13** | **40.64** |

we analyze the trade-off between test accuracy and the resource cost under various sparsity ratios (i.e. $S \in [50\%, 80\%, 90\%]$) and a dense DSN model. Note that the dense model is defined exactly as our DSN model with dense connections along entire kernels. We can see that it is not that the more parameters of the model, the better performance it will achieve. This is mainly because more parameters will lead to overfitting and the proper sparse ratio will easily cover the desired receptive field (more demonstrations can be seen in Appendix B.1). Taking the EEG2 data set as an example, with the decrease of sparse rate, the performance on it drops sharply. We hypothesise that this data set has more expectations for features from small receptive fields (the validation can be found in Appendix B.2). According to the experiments, a 80% sparsity ratio reflects a good trade-off between accuracy and resource cost, which is the default setting for the dynamic sparse CNN layer in our DSN model.

Table 3: Test accuracy (ACC(%)) and resource cost (i.e. Params $(K)\downarrow$ and FLOPs $(M)\downarrow$) of the proposed DSN model with different sparsity ratios on three multivariate TS datasets from UCI.

| Methods | Daily Sport | | | EEG2 | | | HAR | | |
|---|---|---|---|---|---|---|---|---|---|
| | ACC | Params | FLOPs | ACC | Params | FLOPs | ACC | Params | FLOPs |
| Dense | 99.19 | 1059.65 | 262.10 | 89.60 | 1058.11 | 536.17 | 96.06 | 1056.10 | 267.57 |
| $S = 50\%$ | **99.28** | 370.44 | 91.63 | 96.23 | 368.90 | 187.05 | 96.62 | 366.89 | 93.01 |
| $S = 80\%$ | 99.19 | 163.68 | 40.48 | **99.10** | 162.14 | 82.31 | **96.82** | 160.13 | 40.64 |
| $S = 90\%$ | 99.25 | **94.76** | **23.44** | 98.90 | **93.22** | **46.64** | 96.75 | **91.21** | **23.18** |

**Effect of Architecture:** To show how the architecture of DSN affects the final performance, we adopt the critical difference diagram for a detailed comparison, which is usually used for evaluation in TSC [52, 25]. From Figure 5, we can find that the DSN model with 4 dynamic sparse CNN layers and 177 output channels in each layer performs consistently better than the other structures. The proposed model with more output channels seems to be crucial for preserving the capacity of the network, while requires more parameters and FLOPs compared to other counterparts. This highlights an interesting trade-off between accuracy and computational efficiency. According to the experiments, the DSN model with 4 dynamic sparse CNN layers and 141 output channels in each layer is the trade-off between the performance and resource cost, which is the default setting for our DSN model.

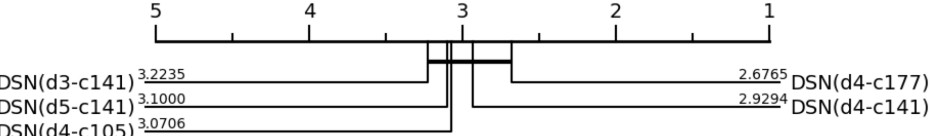

Figure 5: The critical difference diagram shows the average rank of the proposed DSN model with different architectures in UCR 85 archive. DSN($di$-c$j$) represents the DSN model with $i$ dynamic sparse CNN layers and $j$ output channels in each layer. The smaller average rank corresponds to the better performance of the model.

**Effect of Kernel Group:** As shown in Figure 6, training DSN without kernel group hampers the capture of small eNRF, which can not guarantee satisfactory performance on the datasets such as EEG2 expecting more local information. With the decrease in sparsity ratio $S$, the phenomenon that large RF occupies the majority becomes more severe (more demonstrations see in Appendix B.1),

which increases the performance gap between training with group and without it, as shown in Table 4. Grouping the kernels can achieve various RF coverage and adorable accuracy for all datasets.

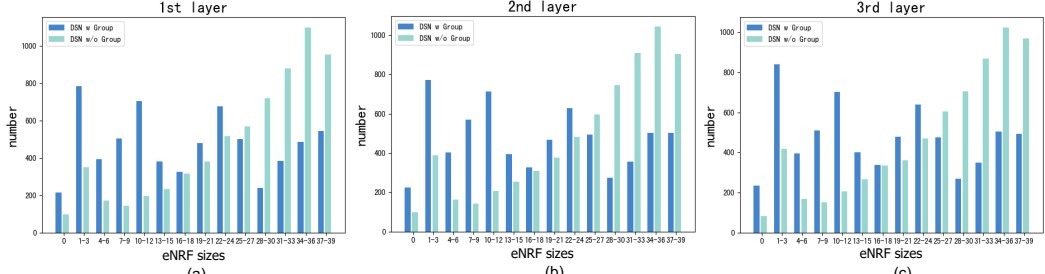

Figure 6: The distribution of eNRF sizes in different sparse CNN layers between DSN models training with and without grouping on HAR dataset. The number on y-axis corresponds to the frequency of eNRF sizes.

## 4.5 Ablation Study

**Effect of Dynamic Sparse Training:** Furthermore, we study the effect of the dynamic sparse training in our DSN method. We conduct ablation studies to compare DSN with its static variant (i.e. $DSN^{fix}$), which fixes its topology during training after activated weights initialization. In other words, the indicator function $\mathbf{I}^l(\cdot)$ in Eq. (1) will not update during the training process. According to the re-

Table 4: Average accuracies (%) of the proposed DSN model affected by kernel grouping.

| Sparsity ratio | Methods | Daily Sport | EEG2 | HAR |
|---|---|---|---|---|
| $S = 60\%$ | DSN w Group | **99.27** | **97.87** | 96.53 |
| | DSN w/o Group | 99.23 | 96.77 | **96.87** |
| $S = 70\%$ | DSN w Group | 99.20 | **98.70** | **97.14** |
| | DSN w/o Group | **99.25** | 97.27 | 96.52 |
| $S = 80\%$ | DSN w Group | 99.19 | **99.10** | **96.82** |
| | DSN w/o Group | **99.28** | 98.77 | 96.79 |

sults shown in Figure 7 (a), we can find that the DSN model mostly outperforms the $DSN^{fix}$ on UCR 85 archive, indicating that what is learned by dynamic sparse training is the suitable activated weights together with their values.

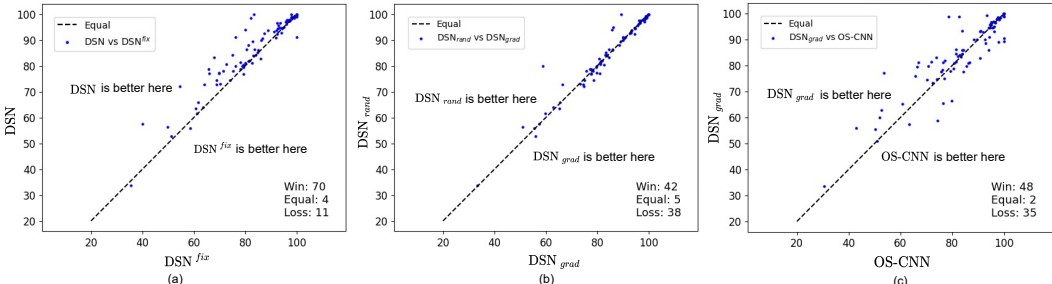

Figure 7: Accuracy plotting showing how the proposed DSN model is affected by the dynamic sparse training in UCR 85 archive.

## 4.6 Case Study on other DST Method

Likewise, the effectiveness of the proposed DSN method is also analyzed in the presence of other DST methods, such as RigL [13]. Rather than growing weights randomly (as in the line 9 of Algorithm 1), RigL grows them with the highest magnitude gradients. From Figure 7 (c), we can see that growing weights according to gradients (i.e. $DSN_{grad}$) can outperforms the OS-CNN method in most cases on

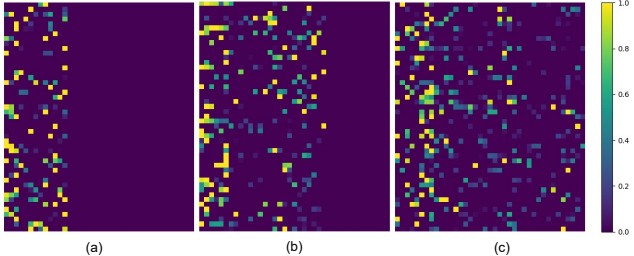

Figure 8: The activated weights in different kernel groups.

UCR 85 archive. Furthermore, as shown in Figure 7 (b), we can find that $DSN_{grad}$ has a similar performance as growing weights randomly (i.e. $DSN_{rand}$). Note that, $DSN_{rand}$ is the main setting of the proposed method in this paper with the reason that we don't need to compute the full gradients at any moment in time. In summary, the proposed DSN with fine-grained sparse strategy can be easily combined with the existing DST techniques, showing the potential for further improvements by merging it with other advanced DST methods in the future.

### 4.7   Visualization of Activated Weights

Figure 8 shows the normalized activated weights of the DSN model after sparse training on HAR dataset. (a), (b) and (c) correspond to the activated weights of the three kernel groups in the first dynamic sparse CNN layer, and each row of the figure represents one kernel. We can see that the weights in each kernel are sparse, and can be activated within the constraint region defined in Section 3.4.

## 5   Conclusions

This paper proposes dynamic sparse network (DSN) to cover diverse effective neighbour receptive field (eNRF) for time series classification (TSC) without the cumbersome hyper-parameters tuning. The proposed DSN can achieve about $2\times$ reduction in memory and computational cost, while achieving superior performance in terms of accuracy on both univariate and multivariate TSC datasets. Furthermore, DSN explores the activated weights in a more fine-grained strategy, which can be easily merged with existing DST techniques, showing a potential for further improvements when combined with advanced technology. DSN provides a feasible solution to bridge the gap between resource awareness and various eNRF coverage for TSC, hoping to inspire other researchers in different fields. It is worthwhile extending DSN to other research domains (e.g. time series forecasting and computer vision). Our DSN model has not yet been executed on real edge devices, and the performance in terms of accuracy and memory footprint on edge devices is not investigated. In the future, we hope to extend our work to other fields and decentralise training on edge devices.

## Acknowledgements

This work has been partly funded by the NWO Perspectief project MEGAMIND. We thank everyone who worked on making the datasets available and their contributions to the TSC community.

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
