# A Preliminaries

## A.1 Preliminaries on Comparison of Exploration Space

The size of exploration space for the layer-wise manner in SET algorithm is:

$$\begin{pmatrix} N_l \\ N_l(1-S) \end{pmatrix} = \frac{N_l!}{N_l(1-S)!N_lS!}. \tag{5}$$

If we divide the kernels in one layer into $N$ groups and explore $\frac{N_l(1-S)}{N}$ activated weights in each group, then the size of the exploration space becomes:

$$\left( \begin{matrix} \frac{N_l}{N} \\ \frac{N_l(1-S)}{N} \end{matrix} \right)^N = \left( \frac{\frac{N_l}{N}!}{\frac{N_l(1-S)}{N}!\frac{N_lS}{N}!} \right)^N \leq \begin{pmatrix} N_l \\ N_l(1-S) \end{pmatrix}. \tag{6}$$

When the corresponding exploration region of each group is defined as in Section 3.4, the exploration space can be further reduced to:

$$\left( \begin{matrix} \frac{N_l}{N^2} \\ \frac{N_l(1-S)}{N} \end{matrix} \right) \left( \begin{matrix} \frac{2N_l}{N^2} \\ \frac{N_l(1-S)}{N} \end{matrix} \right) \cdots \left( \begin{matrix} \frac{N_l}{N} \\ \frac{N_l(1-S)}{N} \end{matrix} \right) < \left( \begin{matrix} \frac{N_l}{N} \\ \frac{N_l(1-S)}{N} \end{matrix} \right)^N. \tag{7}$$

## A.2 Preliminaries on Calculation of Parameters

The summation of the number of non-zero weights (activated weights in sparse CNN layer + weights in other layers) in the network is used to estimate the size of the network as follows:

$$Params = \sum_{l \notin \mathbb{L}} \|\theta^l\|_0 + \sum_{l \in \mathbb{L}} \sum_{i=1}^{N} \|\mathbf{I}^l(\Theta^l)_i \odot \Theta_i^l\|_0 = \|\theta\|_0, \tag{8}$$

where $\mathbb{L}$ contains a set of sparse CNN layer indexes, and $\theta$ is the parameter of our DSN which includes an amount of zero values. $\|\mathbf{I}^l(\Theta^l)_i \odot \Theta_i^l\|_0$ is controlled by the sparsity ratio $S$ and the corresponding exploration regions $\mathbf{R}_i^l$.

# B More Experiments

## B.1 The Statistics of eNRF Sizes with Different Sparsity Ratios

The distributions of eNRF sizes in the first dynamic sparse CNN layer for HAR, Daily Sport and EEG2 datasets are shown in Figure 9, Figure 10 and Figure 11, respectively. We can see that the DSN model with kernel grouping can cover more diverse eNRF sizes than it does without kernel grouping under different sparsity ratios $S$. With the decrease of $S$, the DSN model without kernel grouping tends to capture the large eNRF, while ignoring the small eNRF, which harms the performance of the DSN model. Under different sparsity ratios, the DSN model with kernel grouping can guarantee a satisfactory performance for different datasets by covering various eNRF.

## B.2 Better Kernel Sizes for EEG2 Dataset

We study the effect of kernel size $k$ on the performance of spares/dense DSN for the EEG2 dataset to verify our hypothesis that this dataset expects the local information. The dense DSN model is defined exactly as our DSN model with dense connections along entire kernels. In this way, the RF size in each layer equals to kernel size. As shown in Figure 12, the dense DSN with a kernel size of 9 achieves the highest test accuracy. However, with the increase of kernel size (RF size), the test accuracy drops dramatically. It is obvious that the local context plays a significant role for more accurate classification for EEG2 dataset. In contrast, our proposed sparse DSN model (sparsity ratio $S = 80\%$) with a large kernel size can still achieve satisfactory performance, with the main reason that it can learn the adaptive eNRF for the dataset.

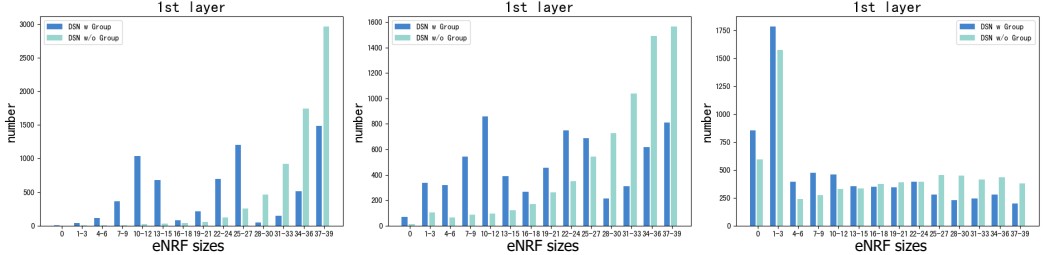

Figure 9: The statistics of eNRF sizes for the HAR dataset. The eNRF sizes shown in a sub-figure are from the first dynamic sparse CNN layer with a specific sparsity ratio (e.g. $S = 50\%$ (left), $S = 70\%$ (middle), and $S = 90\%$ (right)).

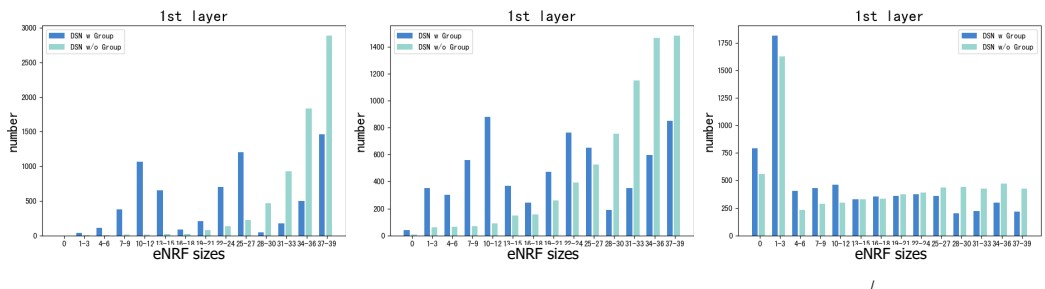

Figure 10: The statistics of eNRF sizes for the Daily Sport dataset. The eNRF sizes shown in a sub-figure are from the first dynamic sparse CNN layer with a specific sparsity ratio (e.g. $S = 50\%$ (left), $S = 70\%$ (middle), and $S = 90\%$ (right)).

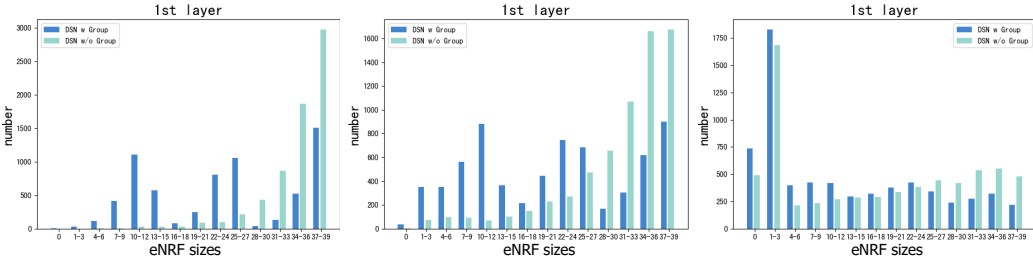

Figure 11: The statistics of eNRF sizes for the EEG2 dataset. The eNRF sizes shown in a sub-figure are from the first dynamic sparse CNN layer with a specific sparsity ratio (e.g. $S = 50\%$ (left), $S = 70\%$ (middle), and $S = 90\%$ (right)).

## B.3 Can Various eNRF Outperform the Optimal NRF?

To demonstrate the advantage of various eNRF coverage, we compare the performances in terms of test accuracy between DSN models with various eNRF coverage and optimal NRF. The dense DSN models with different kernel sizes (from 5 to 40 with step 5) in each layer are used to search for optimal NRF in different datasets on the UCR 85 archive. As shown in Figure 13, by capturing various eNRF, our DSN model can achieve similar performance to the optimal NRF model without the cumbersome hyper parameters.

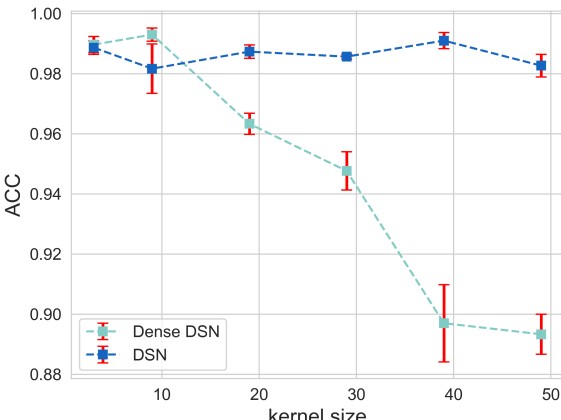

Figure 12: The test accuracies of EGG2 dataset in dense/sparse DSN with different kernel sizes $k$.

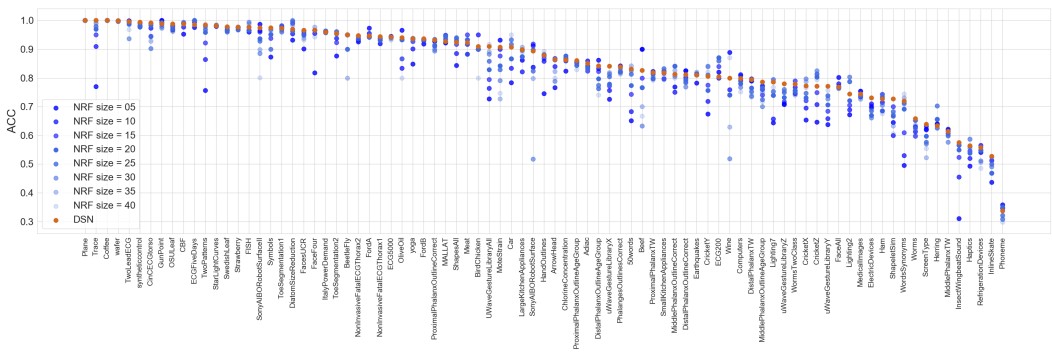

Figure 13: Classification accuracies on UCR 85 archive for the comparison of various eNRF and optimal NRF.

## B.4    The Effect of Kernel Size for DSN

From Figure 14, we can find out how the kernel size $k$ in the proposed DSN model affects the final test accuracy for UCR 85 archive. We can see that the performance of DSN model with small kernel sizes (e.g. 15 and 21) is worse than that of it with large kernel sizes (e.g. 39 and 45). It is because the model with larger kernel sizes can cover larger eNRF, which is useful to capture global information for some datasets. However, when the kernel size is oversized (e.g. 45), some of the small eNRFs can be lost in a predefined sparsity ratio (e.g. $80\%$ in here), which will slightly degrade the performance instead.

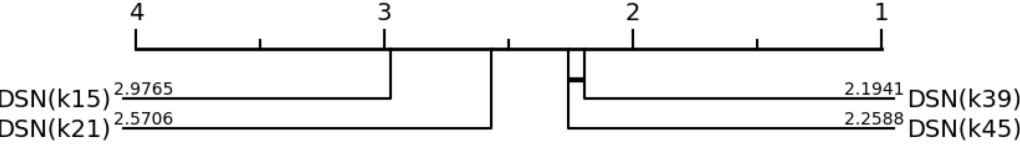

Figure 14: The critical difference diagram shows the average rank of the proposed DSN model with different kernel sizes in UCR 85 archive. DSN($ki$) illustrates that the kernel size in each dynamic sparse CNN layer equals to $i$.

## B.5 The Effect of Kernel Group for DSN

We study how the number of kernel groups $N$ defined in Algorithm 1 affects the performance of the proposed DSN model. From Figure 15, we can see that DSN model with $N = 2$ outperforms other counterparts. As discussed in 4.5 and B.1, grouping the kernels in each dynamic sparse CNN layer can help to achieve various NRF coverage especially with smaller sparsity ratio $S$. However, when the number of kernel groups (e.g. 4) is oversized, the distribution of eNRF sizes is nearly uniform, which will degrade the performance of DSN.

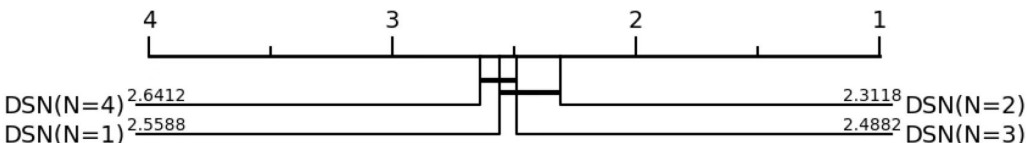

Figure 15: The critical difference diagram shows the average rank of the proposed DSN model with different kernel groups in UCR 85 archive. DSN($N = i$) illustrates that the kernels in each dynamic sparse CNN are split into $i$ groups.

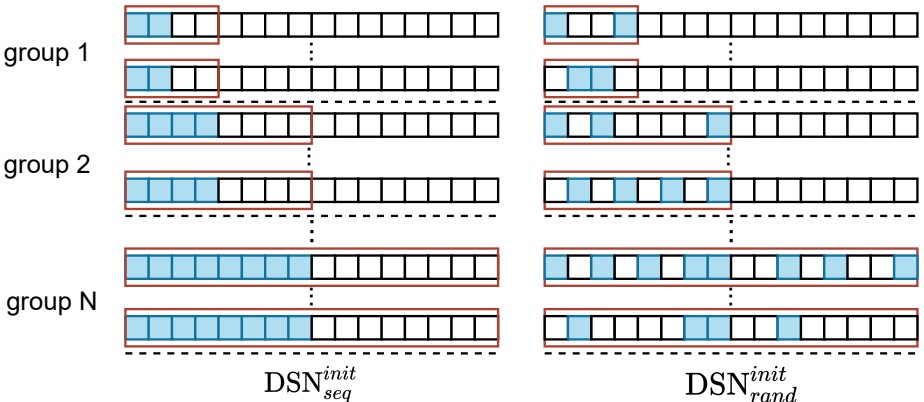

Figure 16: Visualization of different topology initialization manners, including sequential and random. The corresponding exploration regions for each group are shown in the red boxes. The blue color indicates the activated weights at initialization.

## B.6 The Effect of Topology Initialization

As shown in Figure 16, the activated weights can be initialized randomly and sequentially before exploration, namely $DSN_{rand}^{init}$ and $DSN_{seq}^{init}$ respectively. After initialization, the topologies initialized by $DSN_{rand}^{init}$ and $DSN_{seq}^{init}$ are different. In detail, $DSN_{rand}^{init}$ randomly activates the weights within the exploration regions, while $DSN_{seq}^{init}$ activates the first $\frac{i \times k}{N}S$ weights of kernels in $i$th group. Figure 16 exemplifies the case where $k = 16$, $N = 4$ and $S = 50\%$. We further investigate which type of topology initialization is more suitable for the proposed DSN model. According to the results shown in Figure 17 (left) and Figure 18, we can see that the proposed DSN model with sequential initialization performs better than that with random initialization. Based on this, we adopt sequential initialization as the default setting for DSN model in all of the experiments.

Intuitively, the DSN method with random topology can also cover various eNRF. We further study the different performances of NRF which is from dynamic sparse training and from random topology in DSN method. We conduct ablation studies to compare DSN with its static variant (i.e. $DSN_{rand}^{fix}$), which fixes its topology during training after activated weights random initialization. According to the results shown in Figure 17 (right), we can find that the DSN model outperforms the $DSN_{rand}^{fix}$ on UCR 85 archive, indicating that what is learned by dynamic sparse training is the suitable activated weights together with their values.

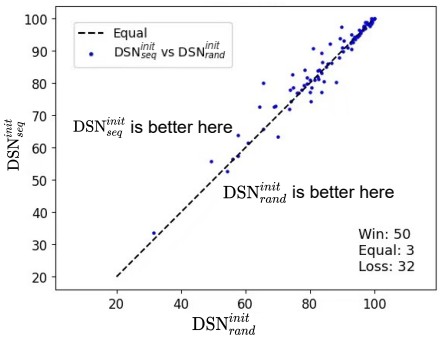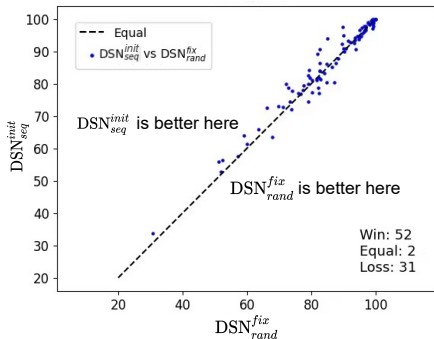

Figure 17: Accuracy plotting showing how the proposed DSN model is affected by the topology initialization on UCR 85 archive.

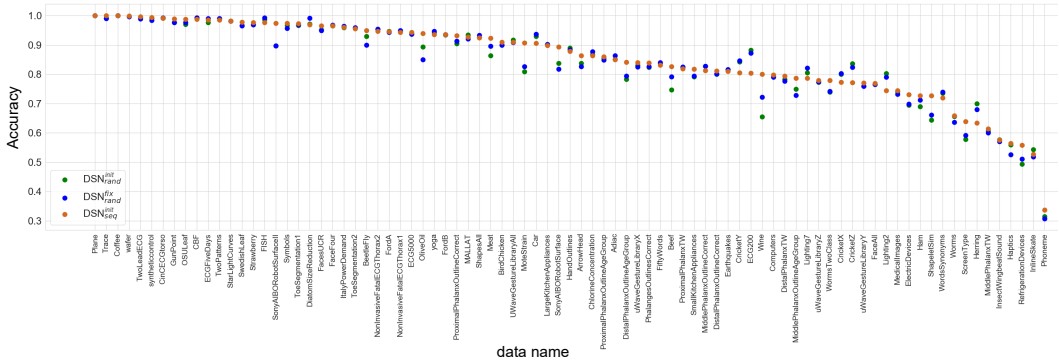

Figure 18: Classification accuracies on UCR 85 archive for the comparison of $\text{DSN}_{seq}^{init}$ and $\text{DSN}_{rand}^{init}$ with/without dynamic exploration.

## B.7  Extremely Sparse DSN

We study how the performance of our DSN is affected under extremely sparse ratios (e.g. $S = 90\%$, $S = 95\%$, and $S = 97\%$). As shown in Table 5, we can see that our DSN with sparsity ratio $S = 95\%$ still performs better than OS-CNN while achieving more than $5\times$ resource cost reduction (e.g. Parameters and FLOPs). When sparsity ratio $S = 97\%$, which means that $97\%$ weights in sparse CNN layer are set to zeros, the performance of our DSN model is only slightly worse than that of OS-CNN, but the resource cost is nearly $7\times$ less. From Figure 19, we can see that most blue dots (corresponding to accuracies of OS-CNN) are below the orange dots (corresponding to accuracies of DSN with extremely sparsity ratio), which means that OS-CNN performs worse than extremely sparse DSN in most datasets. What's more, DSN is far superior to OS-CNN in some datasets (e.g. CinCECGtorso, MiddlePhalanxOutlineAgeGroup, and Earthquakes).

Table 5: Pairwise comparison of test accuracy (%) and mean resource cost (i.e. Params $(K)\downarrow$ and FLOPs $(M)\downarrow$) of our method with extremely sparsity ratio and oS-CNN on UCR 85 archive datasets. Note that all the accuracies in comparison are rounded to two decimal places.

| Archive | Methods | OS-CNN wins | **DSN**$(S)$ wins | Tie | Params | FLOPs |
|---------|---------|-------------|-------------------|-----|--------|-------|
| | OS-CNN | - | - | - | 262.14 | 200.12 |
| | DSN($S = 80\%$) | 33 | **49** | 3 | 126.22 | 104.88 |
| UCR 85 archive | DSN($S = 85\%$) | 33 | **49** | 3 | 100.19 | 83.21 |
| | DSN($S = 90\%$) | 36 | **47** | 2 | 74.16 | 61.54 |
| | DSN($S = 95\%$) | 38 | **43** | 4 | 48.13 | 39.87 |
| | DSN($S = 97\%$) | **41** | 40 | 4 | **37.72** | **31.20** |

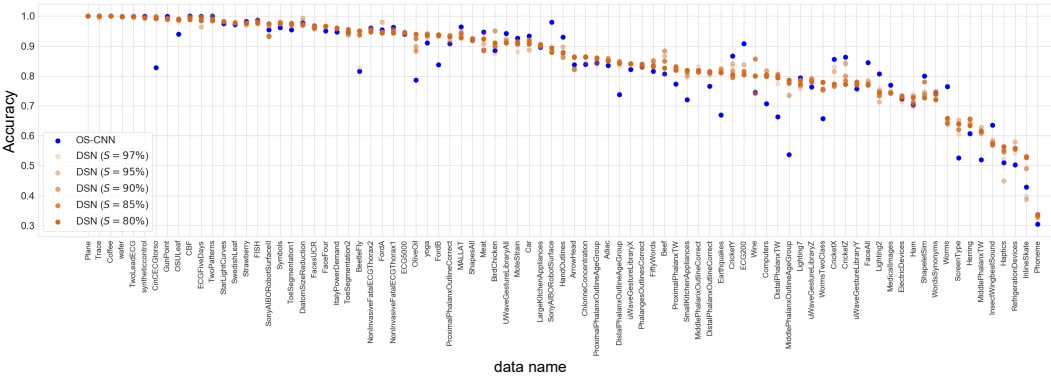

Figure 19: Classification accuracies on UCR 85 archive for DSN with different sparsity ratios.

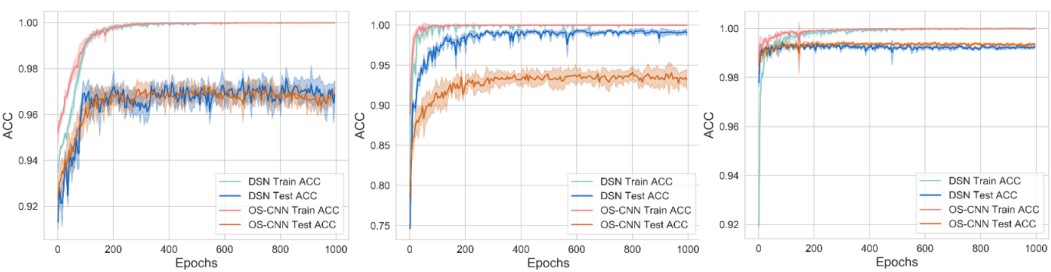

Figure 20: Learning curves of DSN models on different datasets, HAR(left), EEG2(middle), and Daily Sports(right). The standard deviation of the accuracy over 5 runs is shown in the shaded region.

## B.8 Learning Curve on Three Multivariate TS Dataset

Figure 20 shows the learning behaviors of our DSN model and OS-CNN. We can see that our DSN model has a convergence speed similar to that of OS-CNN, while the resource requirements (memory and computational cost) are much less. Furthermore, the performance of DSN is far superior to that of OS-CNN for EEG2 datasets.

## B.9 Additional Experiments on other TSC datasets

In order to evaluate if the good learning capabilities of DSN can be generalized over other larger scale of univariate and multivariate time-series datasets we evaluate it on the following datasets:

- **UCR 2018 archive** [6]. This archive consists of 128 univariate TS datasets, which is the updated version of the UCR 85 archive. Among them, 15 datasets are with unequal lengths and one (Fungi) has a single instance per class in the training files. As shown in [37], most implementations are not set up to handle these kinds of data; therefore, we also only report our results on the remaining 112 problems in UCR 2018 archive.
- **University of East Anglia (UEA) 30 archive** [2]. It consists of 30 multivariate TS datasets with distinguishable characteristics, and various levels of complexity. The class numbers vary from 2 to 39, and their numbers of variates vary from 2 to 963.

For UEA 30 archive, we add the resource awareness method Rocket [7] and the transformer based method TST [61] as our baselines. We summarize the performance on both additional univariate and multivariate TSC benchmarks in Table 6-7. We can see that, in terms of test accuracy, our method can still match the state-of-the-art methods, while having much smaller computational cost (e.g. the number of parameters and FLOPs). It is worth highlighting our proposed DSN method achieves a trade-off between the resource cost and performance in both univariate and multivariate TSC benchmarks.

Table 6: Pairwise comparison of test accuracy (%) and mean resource cost (i.e. Params ($K$)↓ and FLOPs ($M$)↓) of our method and other univariate TS baseline methods on UCR 112 archive datasets. Note that all the accuracies in comparison are rounded to two decimal places.

| Archive | Methods | Baseline wins | DSN wins | Tie | Params | FLOPs |
|---------|---------|---------------|----------|-----|--------|-------|
| UCR 112 archive | ResNet | 43 | **64** | 5 | 479.28 | 1049.12 |
| | InceptionTime | **52** | 51 | 9 | 389.42 | 425.06 |
| | OS-CNN | 48 | **59** | 5 | 264.19 | 259.30 |
| | **DSN (ours)** | - | - | - | **126.30** | **136.91** |

Table 7: Pairwise comparison of test accuracy (%) and mean resource cost (i.e. Params ($K$)↓ and FLOPs ($M$)↓) of our method and other multivariate TS baseline methods on UEA 30 archive datasets. Note that all the accuracies in comparison are rounded to two decimal places.

| Archive | Methods | Baseline wins | DSN wins | Tie | Params | FLOPs |
|---------|---------|---------------|----------|-----|--------|-------|
| UEA 30 archive | TapNet | 11 | **17** | 2 | 1344.67 | 966.46 |
| | MLSTM-FCN | 7 | **23** | 0 | 383.41 | 638.53 |
| | OS-CNN | **15** | 14 | 1 | 389.65 | 547.34 |
| | TST | 12 | **18** | 0 | 343.78 | 628.95 |
| | Rocket | **15** | 14 | 1 | - | 526.87[4] |
| | **DSN (ours)** | - | - | - | **148.74** | **305.37** |

## C  Detailed Results

For UCR 85 archive, the reported results of ResNet and InceptionTime are from [7], while that of OS-CNN are from the official repository [5]. For three multivariate datasets from UCI, the reported results of LSTM-FCN are from the original paper, and that of TapNet and OS-CNN are from the implementation of official repository with default settings. For UEA 30 archive, the detailed results, including test accuracy and resource cost on each dataset, can be found in Table 9.

The detailed results for UCR 85 archive are given in Table 8. We also visualize the detailed results in Figure 21, where we can see that the accuracies of DSN are nearly on the top in most of the datasets. In addition, it is worth noting that DSN loses by a slight margin in some datasets, but wins by a big margin in others.

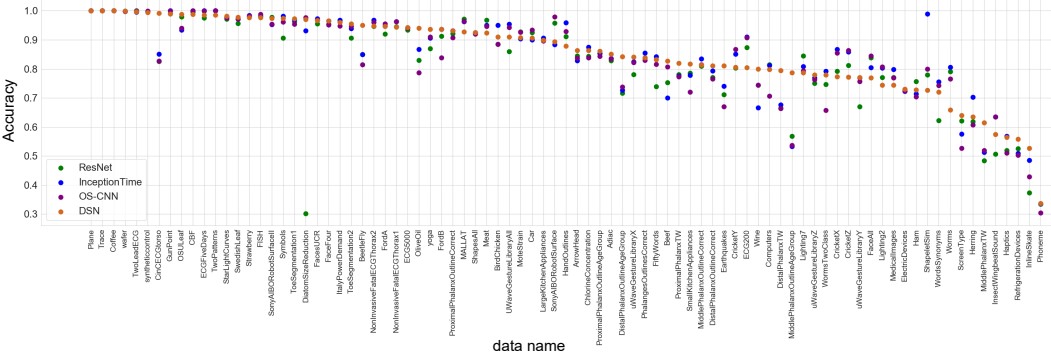

Figure 21: Visualizing classification accuracies on UCR 85 archive for the comparison of DSN and the baselines.

---

[4]For transform operation with 1000 random convolution kernels.
[5]https://github.com/Wensi-Tang/OS-CNN.

# D   More Implementation Details

In our dynamic sparse training algorithm 1, we choose the cosine annealing defined in Eq. 4 with $\Delta T = 5$ epochs and $\alpha = 0.5$. For UEA 30 archive, in each dynamic sparse CNN layer, the output channel is set to 177 with sparsity ratio $S = 90\%$. For UCR 112 archive, except for each CNN layer is padded with zero-value, other hyper-parameters are the same as in UCR 85 archive. We use PyTorch[6] to implement our method and run our experiments on Nvidia Tesla P100.

Table 8: Test accuracies (ACC(%)) for UCR 85 archive and resource cost (i.e. Params (K) and FLOPs (M)) of our method. Test accuracies of our method are run five times and reported with (mean±std).

|  | ResNet | InceptionTime | OS-CNN | DSN (ours) | Params | FLOPs |
|---|---|---|---|---|---|---|
| 50Words | 73.96 | 84.18 | 81.60 | 83.16±0.69 | 132.27 | 68.36 |
| Adiac | 82.89 | 83.63 | 83.45 | 85.06±0.73 | 130.41 | 44.68 |
| ArrowHead | 84.46 | 82.86 | 83.77 | 86.40±0.23 | 125.55 | 62.31 |
| Beef | 75.33 | 70.00 | 80.67 | 82.67±2.49 | 125.83 | 116.65 |
| BeetleFly | 85.00 | 85.00 | 81.50 | 95.00±0.00 | 125.40 | 126.97 |
| BirdChicken | 88.50 | 95.00 | 88.50 | 91.00±2.00 | 125.40 | 126.97 |
| CBF | 99.50 | 99.89 | 99.99 | 98.78±0.31 | 125.55 | 31.82 |
| Car | 92.50 | 90.00 | 93.33 | 90.67±2.26 | 125.69 | 143.14 |
| ChlorineConcentration | 84.36 | 87.53 | 83.87 | 86.33±0.20 | 125.55 | 41.24 |
| CinCECGtorso | 82.61 | 85.14 | 82.75 | 99.20±0.18 | 125.69 | 406.37 |
| Coffee | 100.00 | 100.00 | 100.00 | 100.00±0.00 | 125.40 | 70.95 |
| Computers | 81.48 | 81.20 | 70.68 | 79.84±0.54 | 125.40 | 178.53 |
| CricketX | 79.13 | 86.67 | 85.51 | 77.28±1.57 | 126.83 | 74.71 |
| CricketY | 80.33 | 85.13 | 86.72 | 80.51±1.04 | 126.83 | 74.71 |
| CricketZ | 81.15 | 85.90 | 86.31 | 77.13±1.85 | 126.83 | 74.71 |
| DiatomSizeReduction | 30.13 | 93.14 | 97.71 | 96.99±0.48 | 125.69 | 85.63 |
| DistalPhalanxOutlineAgeGroup | 71.65 | 72.66 | 73.81 | 84.20±0.33 | 125.55 | 19.92 |
| DistalPhalanxOutlineCorrect | 77.10 | 79.35 | 76.59 | 81.13±1.09 | 125.40 | 19.89 |
| DistalPhalanxTW | 66.47 | 67.63 | 66.40 | 79.40±0.37 | 125.98 | 20.00 |
| ECG200 | 87.40 | 91.00 | 90.80 | 80.40±0.49 | 125.40 | 23.85 |
| ECG5000 | 93.42 | 94.09 | 94.01 | 94.32±0.12 | 125.83 | 34.85 |
| ECGFiveDays | 97.48 | 100.00 | 99.95 | 98.51±0.48 | 125.40 | 33.77 |
| Earthquakes | 71.15 | 74.10 | 66.98 | 81.06±0.00 | 125.40 | 126.97 |
| ElectricDevices | 72.91 | 72.27 | 72.36 | 73.02±0.74 | 126.12 | 24.00 |
| FISH | 97.94 | 98.29 | 98.74 | 97.71±0.72 | 126.12 | 114.97 |
| FaceAll | 83.88 | 80.41 | 84.47 | 76.91±0.16 | 127.12 | 32.87 |
| FaceFour | 95.45 | 96.59 | 95.11 | 96.59±0.00 | 125.69 | 86.88 |
| FacesUCR | 95.47 | 97.32 | 96.74 | 96.62±0.23 | 127.12 | 32.87 |
| FordA | 92.05 | 94.83 | 95.48 | 94.64±0.16 | 125.40 | 124.00 |
| FordB | 91.31 | 93.65 | 83.79 | 93.62±0.67 | 125.40 | 124.00 |
| GunPoint | 99.07 | 100.00 | 99.93 | 98.93±0.53 | 125.40 | 37.24 |
| Ham | 75.71 | 71.43 | 70.38 | 72.76±1.66 | 125.40 | 106.89 |
| HandOutlines | 91.11 | 95.95 | 92.95 | 87.90±0.68 | 125.40 | 671.54 |
| Haptics | 51.88 | 56.82 | 51.01 | 56.43±2.30 | 125.83 | 270.82 |
| Herring | 61.88 | 70.31 | 60.78 | 63.44±4.49 | 125.40 | 126.97 |
| InlineSkate | 37.31 | 48.55 | 42.93 | 52.73±4.52 | 126.12 | 466.69 |
| InsectWingbeatSound | 50.65 | 63.48 | 63.52 | 57.49±0.52 | 126.69 | 63.77 |
| ItalyPowerDemand | 96.30 | 96.79 | 94.72 | 96.05±0.13 | 125.40 | 6.01 |
| LargeKitchenAppliances | 89.97 | 90.67 | 89.57 | 89.92±0.26 | 125.55 | 178.56 |
| Lighting2 | 77.05 | 80.33 | 80.66 | 74.43±2.45 | 125.40 | 157.95 |
| Lighting7 | 84.52 | 80.82 | 79.32 | 78.63±1.40 | 126.12 | 79.27 |
| MALLAT | 97.16 | 96.29 | 96.38 | 92.71±1.83 | 126.26 | 254.05 |

---

[6]https://pytorch.org/

| | ResNet | InceptionTime | OS-CNN | DSN (ours) | Params | FLOPs |
|---|---|---|---|---|---|---|
| Meat | 96.83 | 95.00 | 94.67 | 92.33±2.91 | 125.55 | 111.13 |
| MedicalImages | 77.03 | 79.87 | 76.95 | 74.39±0.24 | 126.55 | 24.82 |
| MiddlePhalanxOutlineAgeGroup | 56.88 | 53.25 | 53.64 | 78.65±0.77 | 125.55 | 19.92 |
| MiddlePhalanxOutlineCorrect | 80.89 | 83.51 | 81.41 | 81.33±1.27 | 125.40 | 19.89 |
| MiddlePhalanxTW | 48.44 | 51.30 | 51.95 | 61.45±0.40 | 125.98 | 20.00 |
| MoteStrain | 92.76 | 90.34 | 92.64 | 90.73±0.43 | 125.40 | 20.88 |
| NonInvasiveFatalECGThorax1 | 94.54 | 96.23 | 96.27 | 94.36±0.50 | 131.12 | 187.11 |
| NonInvasiveFatalECGThorax2 | 94.61 | 96.74 | 96.01 | 94.72±0.33 | 131.12 | 187.11 |
| OSULeaf | 97.85 | 93.39 | 94.01 | 98.84±0.31 | 125.98 | 106.01 |
| OliveOil | 83.00 | 86.67 | 78.67 | 94.00±3.27 | 125.69 | 141.40 |
| PhalangesOutlinesCorrect | 83.90 | 85.43 | 82.97 | 83.87±0.78 | 125.40 | 19.89 |
| Phoneme | 33.43 | 33.54 | 30.45 | 33.70±0.79 | 130.70 | 254.93 |
| Plane | 100.00 | 100.00 | 100.00 | 100.00±0.00 | 126.12 | 35.89 |
| ProximalPhalanxOutlineAgeGroup | 85.32 | 85.37 | 84.39 | 86.05±0.50 | 125.55 | 19.92 |
| ProximalPhalanxOutlineCorrect | 92.13 | 93.13 | 90.79 | 93.20±0.14 | 125.40 | 19.89 |
| ProximalPhalanxTW | 78.05 | 77.56 | 77.32 | 81.95±0.29 | 125.98 | 20.00 |
| RefrigerationDevices | 52.53 | 50.93 | 50.29 | 55.84±1.27 | 125.55 | 178.56 |
| ScreenType | 62.16 | 57.60 | 52.64 | 63.95±1.11 | 125.55 | 178.56 |
| ShapeletSim | 77.94 | 98.89 | 79.94 | 72.67±4.87 | 125.40 | 124.00 |
| ShapesAll | 92.13 | 92.50 | 92.03 | 92.50±0.61 | 133.70 | 128.63 |
| SmallKitchenAppliances | 78.61 | 77.87 | 72.08 | 81.76±0.40 | 125.55 | 178.56 |
| SonyAIBORobotSurface | 95.81 | 88.35 | 97.95 | 89.32±1.82 | 125.40 | 17.41 |
| SonyAIBORobotSurfaceII | 97.78 | 95.28 | 95.38 | 97.46±0.75 | 125.40 | 16.17 |
| StarLightCurves | 97.18 | 97.92 | 97.50 | 98.20±0.09 | 125.55 | 253.91 |
| Strawberry | 98.05 | 98.38 | 98.19 | 97.72±0.29 | 125.40 | 58.31 |
| SwedishLeaf | 95.63 | 97.12 | 97.12 | 97.79±0.19 | 127.26 | 32.16 |
| Symbols | 90.64 | 98.19 | 96.12 | 97.43±0.40 | 125.98 | 98.82 |
| ToeSegmentation1 | 96.27 | 96.93 | 95.39 | 97.28±0.33 | 125.40 | 68.72 |
| ToeSegmentation2 | 90.62 | 93.85 | 94.62 | 95.54±0.58 | 125.40 | 85.08 |
| Trace | 100.00 | 100.00 | 100.00 | 100.00±0.00 | 125.69 | 68.28 |
| TwoLeadECG | 100.00 | 99.56 | 99.92 | 99.70±0.07 | 125.40 | 20.38 |
| TwoPatterns | 99.99 | 100.00 | 100.00 | 98.50±0.26 | 125.69 | 31.84 |
| UWaveGestureLibraryAll | 85.95 | 95.45 | 94.25 | 90.99±0.38 | 126.26 | 234.46 |
| Wine | 74.44 | 66.67 | 74.44 | 80.00±2.96 | 125.40 | 58.06 |
| WordsSynonyms | 62.24 | 75.55 | 74.23 | 72.01±1.38 | 128.69 | 67.64 |
| Worms | 79.09 | 80.52 | 76.49 | 65.86±1.83 | 125.83 | 223.23 |
| WormsTwoClass | 74.68 | 79.22 | 65.71 | 77.90±1.35 | 125.40 | 223.15 |
| syntheticcontrol | 99.83 | 99.67 | 99.93 | 99.40±0.13 | 125.98 | 15.05 |
| uWaveGestureLibraryX | 78.05 | 82.47 | 82.18 | 84.06±0.39 | 126.26 | 78.31 |
| uWaveGestureLibraryY | 67.01 | 76.88 | 75.72 | 77.10±0.64 | 126.26 | 78.31 |
| uWaveGestureLibraryZ | 75.01 | 76.97 | 76.36 | 77.98±0.41 | 126.26 | 78.31 |
| wafer | 99.86 | 99.87 | 99.84 | 99.91±0.02 | 125.40 | 37.74 |
| yoga | 87.02 | 90.57 | 91.06 | 93.63±0.48 | 125.40 | 105.65 |

Table 9: Test accuracies (ACC(%)) for UEA 30 archive and resource cost (i.e. Params (K) and FLOPs (M)) of our method. Test accuracies of our method are run five times and reported with (mean±std).

|  | MLSTM-FCN | TapNet | OS-CNN | DSN (ours) | Params | FLOPs |
|---|---|---|---|---|---|---|
| ArticularyWordRecognition | 97.30 | 98.70 | 98.75 | 98.40±0.25 | 146.35 | 41.58 |
| AtrialFibrillation | 26.70 | 33.30 | 23.33 | 6.67±0.00 | 142.00 | 180.42 |
| BasicMotions | 95.00 | 100.00 | 100.00 | 100.00±0.00 | 142.42 | 28.37 |
| CharacterTrajectories | 98.50 | 99.70 | 99.76 | 99.39±0.07 | 145.10 | 52.02 |
| Cricket | 91.70 | 95.80 | 99.31 | 98.89±0.56 | 143.85 | 338.23 |
| DuckDuckGeese | 67.50 | 57.50 | 54.00 | 56.80±3.25 | 221.60 | 119.04 |
| ERing | 13.30 | 13.30 | 88.15 | 92.22±1.12 | 142.66 | 18.55 |
| EigenWorms | 50.40 | 48.90 | 41.41 | 39.08±11.21 | 142.59 | 5075.34 |
| Epilepsy | 76.10 | 97.10 | 98.01 | 99.86±0.29 | 142.24 | 58.21 |
| EthanolConcentration | 37.30 | 32.30 | 24.05 | 24.49±0.89 | 142.24 | 493.68 |
| FaceDetection | 54.50 | 55.60 | 57.50 | 63.49±0.70 | 150.20 | 18.58 |
| FingerMovements | 58.00 | 53.00 | 56.75 | 49.20±1.17 | 143.36 | 14.32 |
| HandMovementDirection | 36.50 | 37.80 | 44.26 | 37.30±2.20 | 142.65 | 113.22 |
| Handwriting | 28.60 | 35.70 | 66.82 | 33.65±0.84 | 146.18 | 43.78 |
| Heartbeat | 66.30 | 75.10 | 48.90 | 78.34±0.79 | 145.30 | 117.00 |
| InsectWingbeat | 16.70 | 20.80 | 66.66 | 38.62±10.65 | 154.94 | 7.07 |
| JapaneseVowels | 97.60 | 96.50 | 99.12 | 98.86±0.11 | 143.66 | 8.53 |
| LSST | 37.30 | 56.80 | 41.25 | 60.26±4.53 | 144.21 | 10.66 |
| Libras | 85.60 | 85.00 | 95.00 | 96.44±0.27 | 144.15 | 13.22 |
| MotorImagery | 51.00 | 59.00 | 53.50 | 57.40±2.58 | 145.48 | 867.23 |
| NATOPS | 88.90 | 93.90 | 96.81 | 97.78±0.93 | 143.84 | 14.72 |
| PEMS-SF | 69.90 | 75.10 | 76.01 | 80.12±1.25 | 199.42 | 57.15 |
| PenDigits | 97.80 | 98.00 | 98.55 | 98.73±0.10 | 143.25 | 2.62 |
| PhonemeSpectra | 11.00 | 17.50 | 29.93 | 31.97±0.36 | 148.98 | 62.77 |
| RacketSports | 80.30 | 86.80 | 87.66 | 86.18±1.32 | 142.42 | 8.61 |
| SelfRegulationSCP1 | 87.40 | 65.20 | 83.53 | 71.74±0.79 | 142.06 | 252.93 |
| SelfRegulationSCP2 | 47.20 | 55.00 | 53.19 | 46.44±4.54 | 142.12 | 325.32 |
| SpokenArabicDigits | 99.00 | 98.30 | 99.66 | 99.10±0.21 | 143.90 | 23.85 |
| StandWalkJump | 6.70 | 40.00 | 38.33 | 38.67±4.99 | 142.12 | 705.04 |
| UWaveGestureLibrary | 89.10 | 89.40 | 92.66 | 91.56±0.44 | 142.95 | 89.08 |