# OpenReview forum: "Dynamic Sparse Network for Time Series Classification: Learning What to “See”"
_NeurIPS.cc/2022/Conference — NeurIPS 2022 Accept_

### Official Review · Reviewer_skdR · 2022-06-17

**Rating:** 5
**Confidence:** 1
**Soundness:** 2 fair
**Presentation:** 3 good
**Contribution:** 2 fair

**Summary:**

The variation of signal scales across and within time series data, makes it challenging to decide proper RF sizes for time series classification. This paper proposes a dynamic sparse network with sparse connections for time series classification, which can learn to cover various receptive field without cumbersome hyper-parameters tuning. The kernels in each sparse layer are sparse and can be explored under the constraint regions by dynamic sparse training, which makes it possible to reduce the resource cost. The experiment results show that the model can achieve state-of-art performance on both univariate and multivariate time series classification datasets.

**Questions:**

1. Whether there are simpler ways to achieve adaptive receptive fields?

2. The significance of experimental results.

3. Are the baselines strong enough?

4. A naive question: can the authors elaborate on how the convolution operators perform between a \in R^k and x^l_i \in R^{hw}? It is not straightforward to me somehow. Why O_j \in R^{hw}?

**Limitations:**

Yes

**Strengths And Weaknesses:**

Strength:
The paper proposes a method that can learn to cover various receptive field without cumbersome hyper-parmeters tuning.

The experiments show that the method achieves state-of-the-art performance on several time series classification datasets.

The paper is well-written and well-structured.

Weakness:

This paper proposes a method that can cover various receptive fields. I wonder whether there are simpler ways to achieve this, e.g. combining the prediction of several models with different receptive fields?

The improvements seem very marginal e.g. on HAR. The authors may need to run the experiments multiple times and report both mean and standard deviation.

I wonder whether the baselines are strong enough? How about comparing to ViT models that has full receptive fields?

---

> ### Author Response · Authors · 2022-08-02
> **Response to Reviewer skdR**
>
> Thanks very much for your valuable comments and questions! Here are our point-wise responses and if you have further questions please feel free to let us know!
>
>
>
> > W1/Q1: This paper proposes a method that can cover various receptive fields. I wonder whether there are simpler ways to achieve this, e.g. combining the prediction of several models with different receptive fields?
>
> A: Thanks for your suggestion. Combining the prediction of several models with different receptive fields could be a solution. However, training several models will increase the resource cost, and require manually configuring the receptive field size for each model. In contrast, our DSN model can adaptively learn to cover  proper eRF by DST during training, while achieving less computational and memory cost.
>
>
>
> > W2/Q2: The improvements seem very marginal e.g. on HAR. The authors may need to run the experiments multiple times and report both mean and standard deviation.
>
> A:  All of our experiments have been repeated for five times and the significance of experimental results have been reported and can be found in Tables 7 and 8 of Appendix.
>
>
>
> > W3/Q3: I wonder whether the baselines are strong enough? How about comparing to ViT models that has full receptive fields?
>
> A: Good point and we agree! We add a state-of-the-art method TST [1] as our baseline and evaluate on UEA 30 multivariate time series benchmark dataset. According to the results shown in the following table, we can see that in terms of test accuracy, our method outperforms the baseline methods in most cases, while achieving more than $3 \times$ resource cost reduction (e.g. Parameters).
>
> | Archive | TST wins | DSN(S=90%) Wins | Tie  | TST Params(K) | DSN(S=90%) Params(K) |
> | ------- | :------: | :-------------: | :--: | :-----------: | :-----------: |
> | UEA 30  |    11    |       13        |  0   |    281.13     |     81.51     |
>
> [1] [Zerveas G, Jayaraman S, Patel D, et al. A transformer-based framework for multivariate time series representation learning. Proceedings of the 27th ACM SIGKDD Conference on Knowledge Discovery & Data Mining. 2021: 2114-2124.](https://dl.acm.org/doi/abs/10.1145/3447548.3467401?casa_token=x8yMpBP-DKsAAAAA:YGqfcZpBCMwwjpGyF31PnUtDVDV1mU5eICedxBcksdNgu5k_mPc2VvpmK9h05QOikR3tTpOS16g)
>
>
>
> > A naive question: can the authors elaborate on how the convolution operators perform between a \in R^k and x^l_i \in R^{hw}? It is not straightforward to me somehow. Why O_j \in R^{hw}?
>
> A:  a \in R^{1\times k} is a one-dimensional kernel/filter that slides through  the input  x^l_i \in R^{h \times w} with the size of h \times w. When stride equals to 1 and padding equals to (k-1)/2, the size of output is equal to size of input (i.e. h \times w ) after the convolution operation.

---

> > ### Comment · Reviewer_skdR · 2022-08-08
> > **Thanks for the response!**
> >
> > Thank the authors for the responses!

---

> ### Author Response · Authors · 2022-08-06
> **Response to Reviewer skdR**
>
> Dear Reviewer skdR:
>
> Thanks a lot for your time and comments! For every concern in your comments, we provided a detailed reply. Would you mind checking them and seeing if they successfully addressed your concerns?
>
> If there is anything unclear or you wish for additional clarification, please let us know soon, so that we can still reply before the author-reviewer discussion period ends. If our response resolves your concerns, we kindly ask you to consider raising the rating of our work. Your support is very important to us and we greatly appreciate that!
>
> Best Regards,
>
> Authors

---

### Official Review · Reviewer_3Hph · 2022-07-08

**Rating:** 6
**Confidence:** 3
**Soundness:** 3 good
**Presentation:** 3 good
**Contribution:** 3 good

**Summary:**

The authors proposed a dynamic sparse network (DSN) with sparse connection to better determine the appropriate receptive field (RF) of time series classification (TSC) tasks. The contribution included alleviating hyper-parameter tuning while maintaining low resource costs. They pointed out the lack of balance between resources and performance without tortuous manual hyperparameter tunings. Where the authors’ DSN model would utilize their designed dynamic sparse CNN layers on resulting effective RF balance with TSC modeling. The experiment was done on real-world time-series datasets and their method showed better accuracy performance as well as lower resource cost

**Questions:**

1. Did they use the same architecture for all Ablation studies, even though each dataset innately carried different characteristics? Is that a fair comparison?

2. In Section 3.4, the authors would need to clarify their weight pruning with more details and insights. Did the weight decay operation train concurrently with the weight activation?

3.   RF was not properly explained properly in the abstract or introduction. “Hidden signals that can be ‘seen’” was not licitly discussed. What is meant by this?

4. The authors could be more specific about their claim of “more fine-grained training strategy” that achieving “diverse effective RF”, so that the readers could better grasp why theirs would be better.

5. The authors could spend a bit more details how the resource awareness would scale up.

6. It would be better for the authors to complete their experiments to have more intuitive illustrations of their model’s impact.


**Limitations:**

The authors have discussed the limitations but only in brief.

**Strengths And Weaknesses:**

The paper has a number of strengths. The authors incorporated dynamic sparse training into RF, which is interesting and could guarantee the region attended to have high information.  Although keeping the kernel sparse during learning was difficult, the authors presented promising results through experiments, the solution could be quite significant for time series work which tends to lag other areas of ML data (images, notes, etc.).

The paper is well written and presented. The authors provided a careful literature review in illustrating the limitations of prior works. The authors’ architecture analysis of both theirs and other works are thorough

The paper has good experimental results. The authors seemed to heuristically apply weight pruning exploration but it was effective, and  provided seemingly comprehensive analysis as they applied 3 different datasets in experiments.

However there are a number of concerns and questions raised about the significance, originality, and technical contribution. The authors would need additional clarification about the implementation of each dataset. Ablation study was done on kernel group or sparse training, but the authors might want to consider taking other components out of the model- Such as sparse CNN modules- and seeing how much each component leads to improvements, to understand the specific contributions of the work. Since they claimed overall contribution was finding the important region in time series data, the authors would need to consider comparing transformer-based works since those would be stronger competitors in sequential data sets (currently none) and attention mechanisms that highlight regions of importance.

Additionally, the authors need clarification on the space of problems solved. Are there bounds on the time-series types of classification problems addressed by this work?

---

> ### Author Response · Authors · 2022-08-02
> **Response to Reviewer 3Hph (1/2)**
>
> Thanks very much for your valuable comments and questions! Here are our point-wise responses and if you have further questions please feel free to let us know.
>
>
>
> > Concern1: The authors would need additional clarification about the implementation of each dataset.
>
> A: We clarify that we do not need to configure a specific hyperparameters for each dataset. All the datasets use the same network architecture and experiment setting, though the number of output channels in conv1x1 layer (shown in Figure 3) should be set as the number of classes for each dataset.
>
>
>
> > Concern2: Ablation study was done on kernel group or sparse training, but the authors might want to consider taking other components out of the model- Such as sparse CNN modules- and seeing how much each component leads to improvements, to understand the specific contributions of the work.
>
> A: Thanks for your suggestion! From Section 4.5, we can find that the components (e.g. kernel group and sparse training) we proposed are important to DSN model. Additionally, we have also verified the effetiveness of sparse CNN modules by comparing with dense modules configured with different kernel sizes (from 5 to 40 with step 5). As shown in Figure 13 (in Appendix),  we can see that  the orange points (accuracy of OS-CNN) are near the top of blue points, that is, our DSN model can achieve comparable performance to the model with optimal RF (from 5 to 40) without the cumbersome hyper parameters.
>
>
>
> > Concern3: The authors would need to consider comparing transformer-based works since those would be stronger competitors in sequential data sets (currently none) and attention mechanisms that highlight regions of importance.
>
> A: Good point and we agree! We add the TST [1] as our baseline on UEA 30 multivariate time series benchmark dataset. The result is as below:
>
> | Archive | TST wins | DSN(S=90%) Wins | Tie  | TST Params(K) | DSN(S=90%) Params(K) |
> | ------- | :------: | :-------------: | :--: | :-----------: | :-----------: |
> | UEA 30  |    11    |       13        |  0   |    281.13     |     81.51     |
>
> We can see that, in terms of test accuracy, our method outperforms the baseline methods in most cases, while achieving more than $3 \times$ resource cost reduction (e.g. Parameters).
>
> [1] [Zerveas G, Jayaraman S, Patel D, et al. A transformer-based framework for multivariate time series representation learning. Proceedings of the 27th ACM SIGKDD Conference on Knowledge Discovery & Data Mining. 2021: 2114-2124.](https://dl.acm.org/doi/abs/10.1145/3447548.3467401?casa_token=67jJ-rcIgr8AAAAA:Y5hrPRlgnu5bhDDsQylbWkf_u7GdcrXYABV-1G4sbLC96rI4WVavC1bf6f7jUlDR94rBg1oJfUs)
>
>
>
> > Concern4: The authors need clarification on the space of problems solved. Are there bounds on the time-series types of classification problems addressed by this work?
>
> A: Dynamic sparse network (DSN) can cover diverse effective receptive field (eRF) without requiring complicated ad-hoc kernel size design. Indeed, it is worthwhile extending DSN to time-series prediction and computer vision domains (e.g. image classification, segementation and detection). In the future, we will extend our work to solve other problems.
>
>
>
> > Question1: Did they use the same architecture for all Ablation studies, even though each dataset innately carried different characteristics? Is that a fair comparison?
>
> A: Yes. In fact, we don't need to configure specific hyperparameters for each dataset for all ablation studies. All the datasets use the same network architecture, though the number of output channels in conv1x1 layer (shown in Figure 3) should be set as the number of classes for each dataset.
>
>
>
> > Question2: In Section 3.4, the authors would need to clarify their weight pruning with more details and insights. Did the weight decay operation train concurrently with the weight activation?
>
> A: Yes, as shown in the 8th line and 9th line of Algorithm 1, $u$ weights in each group will be inactivated. After that, the other $u$ new  weights will be activated. The activated weights  are sparse during the whole training.
>
>
>
> > Question3: RF was not properly explained properly in the abstract or introduction. “Hidden signals that can be ‘seen’” was not licitly discussed. What is meant by this?
>
> A: Thanks for your question.
>
> “Hidden signals that can be ‘seen’” means that the length of time series subsequence used for a unit of feature extraction. We will change the sentence "The receptive field (RF), which determines what hidden signals can be “seen” in a time series model" to "The receptive field (RF), which determines the length of time series to be "seen" and used by a unit of feature extraction".

---

> > ### Author Response · Authors · 2022-08-02
> > **Response to Reviewer 3Hph (2/2)**
> >
> > > Question4: The authors could be more specific about their claim of “more fine-grained training strategy” that achieving “diverse effective RF”, so that the readers could better grasp why theirs would be better.
> >
> > A: Thanks for your suggestion. We have added the following explaination and example in the revision.
> >
> > As shown in Figures 9, 10 and 11in the Appendix, it is hard for  layer-wise DST to capture the small effective RF especially when the sparsity ratio is low (e.g. 50%). For example, when the kernel size is large (e.g. 39 in our case) , capturing the small effective RF (e.g. sizes of 5) means that at least 34 weights to be pruned in the same kernel, which is really hard for the model with a sparsity ratio of 50% (i.e. average pruning 20 weights in a kernel). But, when the channels are divided into 3 different groups, their activated regions are defined and we can guarantee that DSN model covers small (1-13), medium (14-26) and large(27-39) RF, as shown in Figures 9, 10 and 11.
> >
> >
> >
> > > Question5: The authors could spend a bit more details how the resource awareness would scale up.
> >
> > A: We have considered the performance of the DSN model under extremely high sparse ratios in appendix B.7.  As shown in Table 5, our DSN with a sparsity ratio $S = 95\%$ (i.e. 95% weights in sparse CNN layer are set to zeros) still performs better than OS-CNN while achieving more than 5× resource cost reduction (e.g. Parameters and FLOPs). With a sparsity ratio $S = 97%$, the performance of our DSN model is only slightly worse than that of OS-CNN, but the resource cost is nearly 7× reduction.
> >
> >
> >
> > > Question6:  It woubld be better for the authors to complete their experiments to have more intuitive illustrations of their model's impact.
> >
> > A: Thanks for your suggestion. As mentioned above, we have added Minirocket [1] and TST [2] as our new baseline. We can see that, in terms of test accuracy, our method outperforms the baseline methods in most cases. Note that our DSN method which has adaptive receptive field can perform better than TST method with full receptive fields.
> >
> > | Archive | Methods       | Baselines wins | DSN (ours) Wins | Tie  |
> > | ------- | ------------- | :------------: | :-------------: | ---- |
> > | UEA 30  | Minirocket[1] |       10       |        9        | 5    |
> > |         | TST[2]        |       11       |       13        | 0    |
> > |         | DSN (ours)    |       \        |        \        | \    |
> >
> >
> >
> > |               |   Daily Sport   |      EEG2       |       HAR       |
> > | ------------- | :-------------: | :-------------: | :-------------: |
> > | Minirocket[2] | 98.9 $\pm$ 0.05 | 97.6 $\pm$ 0.22 | 97.3 $\pm$ 0.25 |
> > | DSN(ours)     | 99.2 $\pm$ 0.05 | 99.1 $\pm$ 0.30 | 96.8 $\pm$ 0.66 |
> >
> >
> >
> > [1] [Dempster A, Schmidt D F, Webb G I. Minirocket: A very fast (almost) deterministic transform for time series classification. Proceedings of the 27th ACM SIGKDD conference on knowledge discovery & data mining. 2021: 248-257.](https://dl.acm.org/doi/abs/10.1145/3447548.3467231)
> >
> > [2] [Zerveas G, Jayaraman S, Patel D, et al. A transformer-based framework for multivariate time series representation learning. Proceedings of the 27th ACM SIGKDD Conference on Knowledge Discovery & Data Mining. 2021: 2114-2124.](https://dl.acm.org/doi/abs/10.1145/3447548.3467401?casa_token=x8yMpBP-DKsAAAAA:YGqfcZpBCMwwjpGyF31PnUtDVDV1mU5eICedxBcksdNgu5k_mPc2VvpmK9h05QOikR3tTpOS16g)

---

> > > ### Comment · Reviewer_3Hph · 2022-08-05
> > > **Detailed Responses**
> > >
> > > In response - particularly with the ablation study - the authors have improved the results section. Additionally the edits will clarify concerns that were raised.

---

> > > > ### Author Response · Authors · 2022-08-05
> > > > **Thanks for your positive reassessment**
> > > >
> > > > Thanks for your positive reassessment! We are glad to have a very good discussion with you. Please do let us know if there is still anything that you believe we can do to improve it!

---

### Official Review · Reviewer_Ni8e · 2022-07-11

**Rating:** 4
**Confidence:** 4
**Soundness:** 2 fair
**Presentation:** 2 fair
**Contribution:** 1 poor

**Summary:**

The paper proposes DSN, a dynamic sparse network for time-series classification. The goal of DSN is to avoid hyper-parameter tuning and operate with sparse kernels to improve the resources and computational cost. Experiments over univariate and multivariate time-series datasets show that DSN can reduce in half the cost vs. strong recent baselines while achieving state-of-the-art accuracy performance.

**Questions:**

The problem is important. However, there are many concerns with this work:

W1. The paper needs substantial effort to improve its presentation.

RF terminology is not properly introduced. The connection to time-series analysis specifically is never discussed.

There is not enough emphasis on preliminaries and background. Existing compression and sparse DNN solutions are properly introduced. Heavy re-writing is needed

W2. Lack of novelty and technical depth

It's like the first time a sparse NN architecture is applied to the particular time-series classification models, but it's unclear what is novel in this work. This is relevant to the previous comment about properly introducing the 3-4 areas in that space that focus on reducing resources/computational costs. Sparse connection is an emerging area. The paper reads mainly as an adaptation of these solutions and applied to a new domain but the technical depth is not clear, unfortunately.

W3. Missing baselines

How out of the shelf methods that focus on reducing DNN parameters perform on this task. The paper only present how DNS performs vs. baselines that do not consider this problem so for sure there is an improvement. There are multiple works in that area from reducing from FP32 to FP16, from operating with only a few fourier coefficients [a], etc. Multiple baselines in that area are needed to prove how significantly better is DSN

[a] Dziedzic, Adam, et al. "Band-limited training and inference for convolutional neural networks." International Conference on Machine Learning. PMLR, 2019.

W4. UCR contains 128 datasets now and not 85.

There are 30 multivariate time-series datasets in timeseriesclassification.com - All results have to be updated with the recent benchmarks in this area.



**Strengths And Weaknesses:**

Strengths:

S1. Reducing the costs of DNN is a timely and important problem

S2. Extensive experimentation using many public datasets

Weaknesses:

W1. The paper needs substantial effort to improve its presentation.

W2. Lack of novelty and technical depth

W3. Missing baselines

---

> ### Author Response · Authors · 2022-08-02
> **Response to Reviewer Ni8e (1/2)**
>
> Thanks very much for your valuable comments and questions! Here are responses and if you have further questions please feel free to let us know!
>
>
>
> > W1: RF terminology is not properly introduced. The connection to time-series analysis specifically is never discussed.
>
> A: Thanks for pointing out this problem.
>
> * In the revision, we have introduced the Neighbour Receptive Field (NRF), which is defined as the feature region that are used for generating every pixel feature of the successive layer, and distinguished it with RF.
> * As shown in Figure 1 (a),  there are various scaled signals exist  in time series. In the introduction, we claimed that covering as many RF as possible is important in the TSC task, so it does not ignore any useful signals from the time series inputs. In the experiment, we have shown that, by capturing various eRF, our DSN model can achieve comparable performance to the optimal RF model without the cumbersome hyperparameter tuning, as shown in Figure 13.
>
>
>
> > W1: There is not enough emphasis on preliminaries and background. Existing compression and sparse DNN solutions are properly introduced.
>
> A: Thanks for your suggestion!
>
> * We will added a paragraph to explain more on preliminaries and background in the latest version. Indeed, instead of putting too much emphasis on existing compression and sparse DNN solutions, we should make more effort to introduce the dynamic sparse training and highlight the difference between it and other compression/sparsity solutions.
> * We would like to re-emphasize that, unlike other compressed and sparse DNN solutions, DST can dynamically explore (prune and regrow) the activated weights during training. In this way, the exploration of activated weights is of plastic. Additionally, DST makes the network always sparse during the whole training, which can reduce the memory and computation cost during not only the inference process but also the training process.
>
>
>
> > W2. Lack of novelty and technical depth. It's like the first time a sparse NN architecture is applied to the particular time-series classification models, but it's unclear what is novel in this work. This is relevant to the previous comment about properly introducing the 3-4 areas in that space that focus on reducing resources/computational costs. Sparse connection is an emerging area. The paper reads mainly as an adaptation of these solutions and applied to a new domain but the technical depth is not clear, unfortunately.
>
> A3:
>
> * We propose a more fine-grained (group-wise) dynamic sparse training (DST) strategy by grouping channels, which can capture more diverse RF compared with the most used layer-wise DST methods. As shown in Figures 9, 10 and Figure 11, the tiny eRF is hard to be captured by DST methods with a layer-wise exploration manner, especially when the sparsity ratio (e.g. 50%) is undersized. Furthermore, according to Table 4 and Figure 15 , we can see that the fine-grained DST outperformed the layer-wise DST (i.e. without grouping or setting the number of groups to 1) in the univariate and multivariate time series classification. We politely disagree that "the paper reads mainly as an adaptation of these solutions and applied to a new domain".
> * Additionally, we evaluated state-of-the-art transformer-based classifier which has full receptive fields on UEA30 archive. Note that due to limitation of computation power (out of memory during training TST[3]), we can only run 24 dataset in UEA30. We can see that, in terms of test accuracy, our method outperforms the baseline methods in most cases, while achieving more than $3 \times$ resource cost reduction (e.g. Parameters).
>
> | Archive | TST wins | DSN(S=90%) Wins | Tie  | TST Params(K) | DSN(S=90%) Params(K) |
> | ------- | :------: | :-------------: | :--: | :-----------: | :-----------: |
> | UEA 30  |    11    |       13        |  0   |    281.13     |     81.51     |
>
> * As the technical depth,  we have done extensive experiments to verify that the effectiveness of resource-aware DSN model. Our DSN model achieves a satisfactory performance even under extremely high sparse ratios.  Specifically, as shown in Table 5, our DSN with a sparsity ratio $S = 95\%$ (i.e. 95% weights in sparse CNN layer are set to zeros) still performs better than OS-CNN while achieving more than 5× resource cost reduction (e.g, Parameters and FLOPs).

---

> > ### Author Response · Authors · 2022-08-02
> > **Response to Reviewer Ni8e (2/2)**
> >
> > > W3 Missing baselines. How out of the shelf methods that focus on reducing DNN parameters perform on this task. The paper only present how DNS performs vs. baselines that do not consider this problem so for sure there is an improvement. There are multiple works in that area from reducing from FP32 to FP16, from operating with only a few fourier coefficients [a], etc. Multiple baselines in that area are needed to prove how significantly better is DSN
> >
> > A: Thanks for your suggestion! We agree that this is a constructive point to help us further prove the significance of DSN. Thank you pointing out such interesting work [1] that sparsify Fourier coefficients in the frequency domain. However, it is more suitable to add Minirocket [2] which has considered computation cost as our baseline. And the result is as below:
> >
> > |               |   Daily Sport   |      EEG2       |       HAR       |
> > | ------------- | :-------------: | :-------------: | :-------------: |
> > | Minirocket[2] | 98.9 $\pm$ 0.05 | 97.6 $\pm$ 0.22 | 97.3 $\pm$ 0.25 |
> > | DSN(ours)     | 99.2 $\pm$ 0.05 | 99.1 $\pm$ 0.30 | 96.8 $\pm$ 0.66 |
> >
> >
> >
> > [1] [Dziedzic, Adam, et al. "Band-limited training and inference for convolutional neural networks." International Conference on Machine Learning. 2019.](https://proceedings.mlr.press/v97/dziedzic19a.html)
> >
> > [2]  [Dempster A, Schmidt D F, Webb G I. Minirocket: A very fast (almost) deterministic transform for time series classification. Proceedings of the 27th ACM SIGKDD conference on knowledge discovery & data mining. 2021: 248-257.](https://dl.acm.org/doi/abs/10.1145/3447548.3467231)
> >
> > [3] [Zerveas G, Jayaraman S, Patel D, et al. A transformer-based framework for multivariate time series representation learning. Proceedings of the 27th ACM SIGKDD Conference on Knowledge Discovery & Data Mining. 2021: 2114-2124.](https://dl.acm.org/doi/abs/10.1145/3447548.3467401?casa_token=x8yMpBP-DKsAAAAA:YGqfcZpBCMwwjpGyF31PnUtDVDV1mU5eICedxBcksdNgu5k_mPc2VvpmK9h05QOikR3tTpOS16g)
> >
> >
> >
> > > W4 UCR contains 128 datasets now and not 85. There are 30 multivariate time-series datasets in timeseriesclassification.com
> >
> >  A:  Thank you for the information!
> >
> > * As mentioned in the website [1], 15 datasets of  UCR are unequal length and one (Fungi) has a single instance per class in the train files. The official organizer only evaluated state-of-the-art classifiers on the remaining 112 problems. Here, we also only report the result of 112 datasets in UCR in the following table. According to the results, we can see that in terms of test accuracy, our method outperforms the baseline methods in most cases, while achieving more than $2 \times$ resource cost reduction (e.g. Parameters).
> >
> >   | Archive | Methods       | Baselines wins | DSN (ours) Wins | Tie  | Params(K) |
> >   | ------- | ------------- | :------------: | :-------------: | :--: | --------- |
> >   | UCR 112 | ResNet        |       47       |       61        |  4   | 479.20    |
> >   |         | InceptionTime |       52       |       49        |  11  | 389.34    |
> >   |         | OS-CNN        |       47       |       60        |  5   | 262.14    |
> >   |         | DSN (ours)    |       \        |        \        |  \   | 126.22    |
> >
> > * The result of 30 multivariate time-series datasets can be found in appendix B.9. We can find that, in terms of test accuracy, our method outperforms the baseline methods in most cases, while for OS-CNN, our performance can still mostly match with more than $2 \times$ resource cost reduction (e.g. Parameters and FLOPs).

---

> > > ### Comment · Area_Chair_SSHU · 2022-08-09
> > > **Reviewer Ni8e, last call to respond to rebuttal for paper 4394**
> > >
> > > Reviewer Ni8e,
> > >
> > > Today is the deadline to respond to the rebuttal of paper 4394.
> > > Please read the response asap and check whether it addresses your concerns.
> > >
> > > Best regards,
> > >
> > > The AC

---

> ### Author Response · Authors · 2022-08-06
> **Response to Reviewer Ni8e**
>
> Dear Reviewer Ni8e:
>
> Thank you very much for your time and detailed reviews! For every concern in your comments, we provided a detailed reply and added all the mentioned experiments. Since the author-reviewer discussion period has started for a few days, we will appreciate it if you could check our response to your review comments. In this way, if you have further questions and comments, we can still reply before the author-reviewer discussion period ends.
>
> If our response resolves your concerns, we kindly ask you to consider raising the rating of our work. Thank you very much for your time and efforts.
>
> Best Regards,
>
> Authors

---

> ### Author Response · Authors · 2022-08-09
> **[Last day reminder]**
>
> Thanks for the comments! We have tried our best to address all the concerns and updated new experiments to address your concerns. Here is a summary of our response.
>
> * To revise the definition of eRF, we have introduced the **Neighbor Receptive Field (NRF)** and distinguished it with eRF. (**comment 1**)
> * We have added a paragraph to explain more about existing compression and sparse DNN solutions in the appendix E. (**comment 1**)
> * About the novelty, in order to cover diverse and adaptive eNRF, it is the first time to apply dynamic sparse training (DST) strategy to taining CNN model with **large kernel** while reducing computation and memory cost. In addition, compared with traditional dynamic sparse training (DST)  methods, our **fine-grained** (group-wise) DST strategy can capture **more diverse** eNRF and achieve better performance  in time series classification . (**comment 2**)
>
> * As the technical depth, we have done **extensive experiments** to verify the effectiveness of the DSN model on various datasets and shown how the resource awareness would scale up. Our DSN model achieves satisfactory performance even under **extremely high sparse ratios**. (**comment 2**)
> * We have compared our method with the **transformer-based** and another **efficiency awareness** method. DSN also outperforms the baseline methods in most cases.  (**Comment3**)
>
> * As you suggested, we have evaluated our model on another two datasets **UCR112** and  **UEA30**, where DSN indeed shows competitive performance with less computational cost. (**comment 4**)
>
> If there is anything unclear or you wish for additional clarification, please let us know soon, so that we can still reply before the author-reviewer discussion period ends.

---

### Meta-Review · Area_Chair_SSHU · 2022-08-26

**Recommendation:** Accept
**Confidence:** Certain

**Metareview:**

The paper presents a sparse network model for time series classification which dynamically determines receptive fields through the use of sparse kernels. The method matches state of the art performance, while reducing cost.

All the reviewers agreed that the paper solves an important problem and that the authors conducted extensive experiments. There was the question of novelty raised by reviewer Ni8e, although the authors included, in their response, a detailed description of how their work is different from existing methods. The authors also provided clarifications to the issues raised by the other reviewers, even adding new baselines, which were found convincing by reviewers 3Hph and skdR.

**Award:**

No

---

### Decision · Program_Chairs · 2022-09-14

Accept